# Application of simultaneous uncertainty quantification and segmentation for oropharyngeal cancer use-case with Bayesian deep learning

Jaakko Sahlsten [1], Joel Jaskari [1], Kareem A. Wahid [2], Sara Ahmed[2], Enrico Glerean [3], Renjie He[2], Benjamin H. Kann [4], Antti Mäkitie [5,6], Clifton D. Fuller [2], Mohamed A. Naser [2,7] & Kimmo Kaski [1,7]

## Abstract

**Background** Radiotherapy is a core treatment modality for oropharyngeal cancer (OPC), where the primary gross tumor volume (GTVp) is manually segmented with high interobserver variability. This calls for reliable and trustworthy automated tools in clinician workflow. Therefore, accurate uncertainty quantification and its downstream utilization is critical.

**Methods** Here we propose uncertainty-aware deep learning for OPC GTVp segmentation, and illustrate the utility of uncertainty in multiple applications. We examine two Bayesian deep learning (BDL) models and eight uncertainty measures, and utilize a large multi-institute dataset of 292 PET/CT scans to systematically analyze our approach.

**Results** We show that our uncertainty-based approach accurately predicts the quality of the deep learning segmentation in 86.6% of cases, identifies low performance cases for semi-automated correction, and visualizes regions of the scans where the segmentations likely fail.

**Conclusions** Our BDL-based analysis provides a first-step towards more widespread implementation of uncertainty quantification in OPC GTVp segmentation.

## Plain language summary

Radiotherapy is used as a treatment for people with oropharyngeal cancer. It is important to distinguish the areas where cancer is present so the radiotherapy treatment can be targeted at the cancer. Computational methods based on artificial intelligence can automate this task but need to be able to distinguish areas where it is unclear whether cancer is present. In this study we compare these computational methods that are able to highlight areas where it is unclear whether or not cancer is present. Our approach accurately predicts how well these areas are distinguished by the models. Our results could be applied to improve the computational methods used during radiotherapy treatment. This could enable more targeted treatment to be used in the future, which could result in better outcomes for people with oropharyngeal cancer.

Management of oropharyngeal cancer (OPC), a type of head and neck squamous cell carcinoma (HNSCC), still remains a challenge even for experienced multidisciplinary centers[1]. A core treatment modality in OPC patient care is radiotherapy (RT). The current standard of care for OPC RT relies on clinical experts' manually generated segmentation of the primary gross tumor volume (GTVp) as a target structure to deliver RT dose. However, the GTVp in OPC is notorious for being one of the most difficult structures amongst all cancer types to perform accurate segmentation for RT planning due to its exceptionally high interobserver variability[2–4]. Subsequently, GTVp segmentation has been cited as the single largest factor of

[1]Department of Computer Science, Aalto University School of Science, Espoo, Finland. [2]Department of Radiation Oncology, The University of Texas MD Anderson Cancer Center, Houston, TX, USA. [3]Department of Neuroscience and Biomedical Engineering, Aalto University School of Science, Espoo, Finland. [4]Artificial Intelligence in Medicine Program, Brigham and Women's Hospital, Dana-Farber Cancer Institute, Harvard Medical School, Boston, MA, USA. [5]Department of Otorhinolaryngology, Head and Neck Surgery, University of Helsinki and Helsinki University Hospital, Helsinki, Finland. [6]Research Program in Systems Oncology, University of Helsinki, Helsinki, Finland. [7]These authors jointly supervised this work: Mohamed A. Naser, Kimmo Kaski. ✉e-mail: manaser@mdanderson.org; kimmo.kaski@aalto.fi

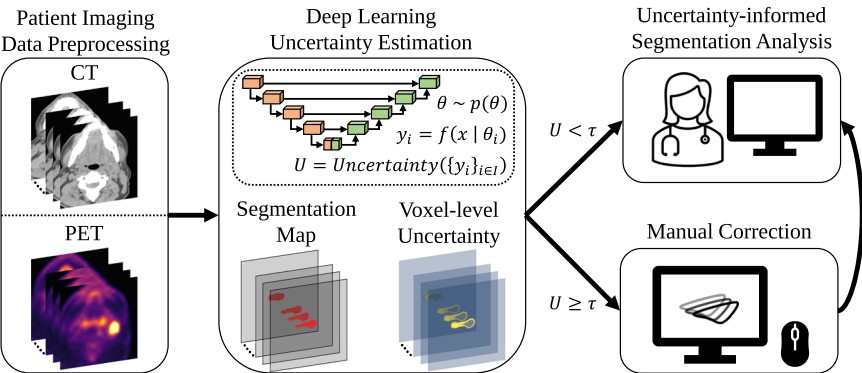

**Fig. 1 | Proposed framework for uncertainty-aware GTVp segmentation of OPC patients.** The probabilistic deep learning model ($f(\cdot)$) with stochastic parameters ($\theta$) distributed according to an approximate posterior distribution ($p$) segments the GTVp, outputs a voxel-level uncertainty map, and quantifies the patient-level uncertainty value ($U$). The patient-level uncertainty is then used to estimate the segmentation quality by checking whether the uncertainty is below or above the predetermined threshold ($\tau$). When the patient-level uncertainty exceeds the threshold, a medical expert will manually inspect and perform corrections to the deep learning segmentation, if necessary. The downstream utilization of the segmentation is then informed by the patient-wise and voxel-wise uncertainties, as well as the patient-wise performance estimate.

uncertainty in RT planning[5,6]. Therefore, automated approaches that can reduce interobserver variability are of paramount importance in improving the current OPC RT workflow.

Deep learning (DL) has increasingly been used in the OPC RT space to automatically segment organs at risk[7,8] and target structures[9–12]. Impressively even for GTVp segmentation, several DL approaches have boasted exceptionally high performance in terms of volumetric and surface-level agreement with the ground-truth segmentations[13]. Importantly, the DL-based auto-segmentation has been shown to be superior to the expected expert's interobserver variability[14], thereby highlighting its potential utility as a support tool for accurate clinical decision-making. Notably, many of these advances in OPC GTVp segmentation have been spurred by open-source data challenges[15], namely the HEad and neCK TumOR (HECKTOR) PET/CT tumor segmentation challenge[14,16,17]. However, while there exists a deluge of DL-based OPC auto-segmentation approaches that demonstrate potentially clinically acceptable performance in terms of geometric measures as evident from HECKTOR[16], the relative confidence (i.e., uncertainty) with which the predictions generated by these models remains a relatively unexplored domain.

Uncertainty quantification is crucial to improve the trust of clinicians in automated systems and to facilitate the clinical implementation of these technologies[18]. Within RT, the segmentation is a clear and well-discussed application space for uncertainty estimation[19]. This is particularly relevant for RT target structures (i.e., OPC GTVp) where high interobserver variability is expected. In addition, there is interest in separating the predictive uncertainty into aleatoric and epistemic components and analyzing them separately[20]. While the performance of OPC GTVp auto-segmentation models is seemingly impressive, the actual clinical utility for most of these methods has yet to be solidified due to a lack of investigations on model uncertainty. Previous work in DL uncertainty estimation has been extensively investigated in segmenting lung-related[21–23] and brain-related[24–26] structures. While DL uncertainty estimation has been applied to a broad range of HNSCC-related classification tasks[27–30] and dose prediction[31], only a limited number of studies have investigated uncertainty estimation for 3-dimensional HNSCC medical image segmentation, predominantly for nasopharyngeal cancer[32] or organs at risk[33,34]; to our knowledge only one study has attempted to investigate segmentation uncertainty estimation in OPC[35]. Therefore, there exists a considerable gap in knowledge about how to construct DL auto-segmentation models that lend themselves to uncertainty estimation and subsequently how to quantify the model uncertainty at individual patient and voxel-wise levels for OPC GTVp segmentation.

In this study, we explore the utilization of uncertainty with deep learning in 3D medical imaging by proposing an uncertainty-aware DL for the OPC GTVp auto-segmentation. We develop probabilistic DL models based on Deep Ensemble and MC Dropout Ensemble using large-scale PET/CT datasets and systematically investigate various established uncertainty measures for patient-level uncertainty, i.e., the information-theoretic entropy, expected entropy, mutual information, coefficient of variation, and structure expected entropy. Here we contribute by deriving three novel measures to our best knowledge, namely the Dice-risk, structure entropy, and structure mutual information and present a qualitative analysis of voxel-wise uncertainty with the entropy, expected entropy, and mutual information. We evaluate the auto-segmentation results with established segmentation performance measures, and we evaluate the utility of uncertainty information by employing several quantitative evaluation methods to link uncertainty measures to the known performance measures and qualitatively investigate the results of uncertainty.

## Methods

This research is based on retrospective and registry-based data, and as such does not consider human subjects and is not subject to IRB approval. Our external validation dataset was retrospectively collected under a HIPAA-compliant protocol approved by the MD Anderson institutional Review Board (RCR03-0800) which implements a waiver of informed consent. In this section, we introduce background on uncertainty estimation in deep learning, present the materials and experimental setup, i.e., the datasets used in this study, describe the DL models we employ, introduce the uncertainty measures that are used to quantify the model uncertainty, and list all the performance evaluation metrics used for the experiments. Our proposed uncertainty-based framework is depicted in Fig. 1.

### Uncertainty estimation in deep learning

Approaches based on conventional DL have been found to be overconfident in the predictions they make. This means that the probability estimates they provide do not correspond to the observed likelihood of them being correct[36]. The Bayesian approach has been described to show promise in improving uncertainty estimation and calibration of DL methods[37]. A direct estimation of the Bayesian posterior of the parameters is intractable, and thus approximation methods are required. Two common posterior approximation methods for segmentation in the literature are deep ensembling[22,38] and Monte Carlo (MC) Dropout[26,39,40]. The deep ensemble approach assumes that multiple networks, having been trained independently with stochastic gradient descent with weight decay, provide samples from the modes of the posterior. The MC Dropout can be seen as a variational approximation of the posterior[41] that can be efficiently sampled from. A key concept for the Bayesian predictive models is the posterior predictive distribution that requires marginalization over the model parameters. Using samples from the deep ensemble or MC Dropout, we can use Monte Carlo

approximation for the posterior predictive distribution:

$$p(y|x) = E_{\theta \in \Theta}[p(y|x, \theta)] \approx \frac{1}{M} \sum_{(m=1)}^{M} p(y|x, \theta^{(m)}), \quad (1)$$

where $M$ is the number of MC samples and $\theta^{(m)}$ is the m:th MC sample of the parameters.

Uncertainty in any modeling task is often thought to be separable to two main sources i.e., being of aleatoric or epistemic origin[42]. The aleatoric uncertainty is the component that is caused by inherent randomness in the data, e.g., noise, while the epistemic uncertainty originates from imperfect knowledge of the model, e.g., the type of model or its parameters. In deep learning, both aleatoric and epistemic uncertainty cause predictive uncertainty that is the uncertainty that the deep learning model has in its predictions[43].

To measure the uncertainty associated with a set of events, a classic uncertainty measure is the information-theoretic entropy, proposed by Shannon[44], defined for a predictive model of $y$ given $x$ as:

$$H[p(y|x)] = -\sum_{y \in Y} p(y|x) \log[p(y|x)] = -E_{y \in Y}[\log[p(y|x)]]. \quad (2)$$

For a Bayesian supervised predictive model with the probability distribution over the set of possible parameters $\Theta$, such as a Bayesian neural network (BNN), the predictive entropy of the model can be decomposed to the aleatoric and epistemic components[45] as follows:

$$H[p(y|x)] = H[E_{\theta \in \Theta}[p(y|x, \theta)]] = E_{\theta \in \Theta}[H[p(y|x, \theta)]] + I[y; \Theta], \quad (3)$$

where $\theta$ is a realization (or an event) of the parameters, $I[\cdot; \cdot]$ is the mutual information between the output of the model and the parameters of the model, and the term $E_{\theta \in \Theta}[H[p(y|x, \theta)]]$ captures the aleatoric uncertainty while $I[y; \Theta]$ stands for the epistemic uncertainty component. For the remainder of this work, $H[p(y|x)]$ will be called the predictive entropy and denoted with $H$, $E_{\theta \in \Theta}[H[p(y|x, \theta)]]$ will be called the expected entropy and denoted with $EH$, and the mutual information will be denoted with $I$. A common approach in deep learning segmentation is to consider that the labels of each voxel are conditionally independent of each other given the input, see e.g., refs. 46,[47]. Hence the entropy-based uncertainty of the output as a whole is the sum of the voxel-wise entropies.

Recently, there has also been interest in defining alternative uncertainty measures for deep learning. One approach to such measures is to view the uncertainty in terms of pointwise risk in $x$[48–50]. That is:

$$R(x) = E_{y \in Y}[c(y, f(x))], \quad (4)$$

where $c(\cdot, \cdot)$ is a cost function and $f(x)$ is the output of a neural network. When the cost function is negative log-likelihood and assume that the output of the neural network is the true conditional distribution of possible events, we have:

$$R_{NLL}(x) = E_{y \in Y}[-\log[p(y|x)]] = H[p(y|x)]. \quad (5)$$

Thus, the information-theoretic entropy can be viewed as a special case of the pointwise risk with a negative log-likelihood as the cost function. Furthermore, the risk-based approach allows one to design a wide variety of uncertainty measures depending on how one views what the cost should be in the setting. However, since we do not necessarily have a cost function that is of an additive form of voxel-wise costs, the voxel-wise uncertainty might not be computable. In addition, for real life use-cases, it requires special modeling tools to decompose the risk-based uncertainty into aleatoric and epistemic components[50].

In addition to the entropy and risk-based view to uncertainty estimation, there have been numerous ad hoc uncertainty measures proposed

for medical DL segmentation that have not been derived from the first principles, but have achieved remarkable results in the tasks involved. For example, Roy et al.[26] proposed so-called structure-wise uncertainty computation, where the uncertainty is only computed on voxels that the model deems as part of the foreground structure, e.g., in GTVp segmentation the voxels where $p(y_{i,j,k} = GTVp|x) \geq 0.5$. In the study, the structure predictions were then used to calculate two novel uncertainty measures, i.e., the coefficient of variation of the volume of thresholded predictions and the expected entropy of the structure voxels. In addition, the work proposed a third novel uncertainty measure called pairwise Dice similarity coefficient which was not included in our study due to the high computational cost when using a large number of MC samples combined with ensembling.

Since the theory of uncertainty estimation with deep learning is still a developing field, and there does not exist a universal approach to it, systematic evaluation of uncertainty-aware neural networks and uncertainty measures for each use-case is necessary.

## Materials and experimental setup

In this section, we present the dataset and the experimental setup used for our results.

**Dataset**. For this study, we utilized two main OPC patient datasets containing PET/CT data: (1) the publicly available 2021 HECKTOR Challenge training dataset[51], which we obtained by completing the End User Agreement through AICrowd[52], and (2) an external validation dataset from The University of Texas MD Anderson Cancer Center (MDA). The HECKTOR dataset contains 224 OPC patients with co-registered PET/CT scans. Each scan included a GTVp segmentation mask manually generated by a single clinical annotator and multiple annotators were involved for the whole dataset. Additional details on the HECKTOR dataset can be found in the corresponding overview paper[51]. The MDA external validation dataset contains 67 human papilomavirus-positive OPC patients with co-registered PET/CT scans with manually generated GTVp segmentation masks from a single clinician annotator (S.A.). Manual segmentation was performed using Velocity AI software v. 3.0.1 (Atlanta, GA, USA). Additional details on the MDA external validation dataset, including image acquisition characteristics and demographic variables, can be found in Supplementary Methods, Supplementary Tables 1 and 2. The MDA external validation dataset was retrospectively collected under a HIPAA-compliant protocol approved by the MDA institutional review board (RCR03-0800) which implements a waiver of informed consent.

For model training and evaluation, all data was resampled into 1 mm isotropic pixel spacing, 1-mm slice thickness, and cropped into $144 \times 144 \times 144$ voxel-sized volumes centered around the GTVp segmentation. The CT scans were windowed at $[-200, 200]$ Hounsfield Units and rescaled to $[-1, 1]$ range, and the PET scans were z-score normalized. The models were trained using a fivefold cross-validation scheme on the HECKTOR dataset. For the performance evaluation of the model, the MDA external validation dataset was used.

**Bayesian deep learning models**. We investigated two approximations of Bayesian inference in DL, i.e., the Deep Ensemble and the Monte Carlo (MC) Dropout Ensemble. The DL architecture in both models used the same 3D residual U-net from the Medical Open Network for AI (MONAI) (0.7.0)[53] that was chosen due to its established success in OPC GTVp segmentation[10,12,51,54]. This architecture has two input channels, i.e., one for CT and one for PET, and a single-output channel with the sigmoid activation function. The input is followed by an encoder consisting of five convolution blocks with 16, 32, 64, 128, and 256 channels, followed by a decoder mirroring the channel count, and a feature concatenation from the decoder to the respective encoder block. Each of these blocks has two convolution layers each followed by instance normalization, dropout, and parametric ReLU layers, and a residual connection with convolution between the input and output of the block.

The only difference between the methods is that the MC Dropout method applies the dropout stochastic regularization layer during test-time[41], whereas the output was deterministic with the Deep Ensemble.

Both ensembles consisted of five models that were each trained using fivefold cross-validation for the HECKTOR dataset. From a Bayesian point of view, the posterior predictive distribution of a Deep Ensemble is approximated with a uniform mixture of the individual networks in the ensemble, i.e., the average of the predictions of the individual networks is approximate Bayesian inference. With the MC Dropout Ensemble, the uniform mixture is over multiple networks with MC dropout, and the predictive distribution is approximated by the average over Monte Carlo samples from each of these networks[55]. In practice, we used 60 MC samples from each of the five ensemble members for the approximation.

For both ensembles, the optimal hyperparameters i.e., loss functions and dropout rate were searched based on the cross-validation performance. For the loss function, we evaluated the Dice loss, which is a soft approximation to the Dice similarity coefficient:

$$DiceLoss(y, f(x)) = 1 - 2 \frac{\Sigma_{i,j,k} y_{i,j,k} f(x)_{i,j,k}}{\Sigma_{i,j,k} y_{i,j,k} + \Sigma_{i,j,k} f(x)_{i,j,k}}, \qquad (6)$$

where $y_{i,j,k}$ and $f(x)_{i,j,k}$ are the label and prediction values at the coordinate $(i, j, k)$, respectively. In addition, we evaluated the sum of Dice and Binary Cross-Entropy (BCE) losses:

$$BCELoss(y, p(y|x)) = - \sum_{i,j,k} (y_{i,j,k} \log[f(x)_{i,j,k}] \\ + (1 - y_{i,j,k}) \log[1 - f(x)_{i,j,k}]), \qquad (7)$$

$$DiceBCELoss(y, p(y|x)) = DiceLoss(y, f(x)) + BCELoss(y, f(x)), \qquad (8)$$

where $f(x) = p(y = GTVp|x, \theta)$. The dropout rate was evaluated with values of 0.1, 0.2, …, 0.9. The tuning of the model was based on the combination of the overall segmentation performance and the quality of the uncertainty estimation with the largest value for area under the Dice similarity coefficient referral curve described in detail in the Uncertainty measures subsection. To keep our analysis as general as possible, uncertainty was not utilized otherwise during model development or training. The final model for both the Deep Ensemble and MC Dropout Ensemble utilized a dropout rate of 0.5 and the DiceBCELoss.

**Segmentation performance evaluation**. The predictions of the models $p(y_{i,j,k}|x)$ were thresholded with values of 0.5 and higher being considered as GTVp and otherwise as background. We denote the thresholded predictions with $\widetilde{y}$ and the binary mask with $y$. We evaluated the segmentation performance with the Dice similarity coefficient (DSC):

$$DSC(y, \widetilde{y}) = 2 \frac{\Sigma_{i,j,k} y_{i,j,k} \widetilde{y}_{i,j,k}}{\Sigma_{i,j,k} y_{i,j,k} + \Sigma_{i,j,k} \widetilde{y}_{i,j,k}}, \qquad (9)$$

the mean surface distance (MSD):

$$MSD(y, \widetilde{y}) = \frac{1}{2} \left( \frac{1}{|\partial \widetilde{y}|} \Sigma_{a \in \partial \widetilde{y}} min_{b \in \partial y} ||a - b||_2 + \frac{1}{|\partial y|} \Sigma_{a \in \partial y} min_{b \in \partial \widetilde{y}} ||a - b||_2 \right), \qquad (10)$$

and the mean Hausdorff distance at 95% (95HD):

$$95HD(y, \widetilde{y}) = max\{ max_{P_{95}} \{ min_{b \in \partial y} ||a - b||_2 : a \in \partial \widetilde{y} \}, \\ max_{P_{95}} \{ min_{b \in \partial \widetilde{y}} ||a - b||_2 : a \in \partial y \} \}, \qquad (11)$$

where $\partial$ is an operator that extracts the set of surface voxels, $|\cdot|$ is the the cardinality of a set, and $max_{P_{95}}$ is the 95th percentile maximum. These

metrics were selected because of their ubiquity in the literature and ability to capture both volumetric overlap and boundary distances[56,57]. For the fivefold cross-validation results, we report the mean and standard error of the mean (SEM) of the metrics computed on each fold, whereas for the holdout set we report the mean and interquartile range (IQR) of point estimates. The model output was resampled into original resolution with nearest-neighbor sampling and evaluated against original resolution segmentations. The performance of MSD and 95HD was evaluated in millimeters. When comparing the segmentation model metrics, we implemented two-sided Wilcoxon signed-rank tests with P values less than or equal to 0.05 considered as significant. Statistical comparisons and their annotations on figures were performed using the SciPy (1.7.3)[58] and statannotations (0.4.4)[59] Python packages, respectively.

**Uncertainty measures**. We consider predictive entropy $H$, expected entropy $EH$, and mutual information $I$ as the entropy-based uncertainty measures. We also examine the risk-based uncertainty estimation, for which we propose negative $DSC$ to be the cost function in GTVp segmentation. We call this uncertainty measure DSC-risk ($R_{DSC}$), which is calculated as:

$$R_{DSC}(x) = E_{y \in Y}[-DSC(y, \widetilde{y})], \qquad (12)$$

where the expectation over $y$ is with respect to the posterior predictive distribution $p(y|x)$. Since the expectation over all the possible segmentations $Y$ cannot be calculated in practice, we utilize instead a Monte Carlo estimate of the DSC-risk:

$$R_{DSC}(x) \approx - \frac{1}{M} \sum_{(m=1)}^{M} DSC(c_m, \widetilde{y}), c_m \sim p(y = GTVp|x). \qquad (13)$$

Thus, this estimate is taken in "doubly stochastic" manner, since $p(y|x)$ is also estimated with Monte Carlo approximation. We decided not to consider the MSD and 95HD-based cost functions with the risk-based uncertainty estimation in order to keep the number of uncertainty measures within reasonable limits.

In addition, we use the recently proposed structure-based uncertainty estimates i.e., coefficient of variation ($CV$) and structure expected entropy ($SEH$), which are defined next[26]. Let $G(x, \theta)$ be the set of voxels that the model with parameters $\theta$ predicts as part of the GTVp structure:

$$G(x, \theta) = \{(i, j, k) : p(y_{i,j,k} = GTVp|x, \theta) \geq 0.5\}. \qquad (14)$$

Then the $CV$ and $SEH$ are calculated as:

$$CV(x) = \frac{\sqrt{E_{\theta \in \Theta}[|G(x,\theta)|^2] - E_{\theta \in \Theta}[|G(x,\theta)|]^2}}{E_{\theta \in \Theta}[|G(x,\theta)|]} = \frac{\sigma_g}{\mu_g}, \qquad (15)$$

$$SEH(x) = E_{\theta \in \Theta} \left[ \frac{1}{|G(x,\theta)|} \Sigma_{g \in G(x,\theta)} H[p(y_g|x, \theta)] \right], \qquad (16)$$

where $|\cdot|$ stands for the cardinality of a set i.e., the number of voxels predicted as GTVp, $\sigma_g$ the standard deviation, and $\mu_g$ the mean of the number of voxels predicted as GTVp with respect to the parameter distribution of the deep learning model. The expectations over the model parameters are computed with Monte Carlo approximation.

To study the structure uncertainty with entropy-based measures in depth, we also have a minor contribution of combining the two other entropy-based measures with structure uncertainty calculation. Hence we define the structure predictive entropy ($SH$) and structure mutual information ($SI$) as follows:

$$G(x) = \{(i, j, k) : p(y_{i,j,k} = GTVp|x) \geq 0.5\}, \qquad (17)$$

$$SH(x) = \frac{1}{|G(x)|} \Sigma_{g \in G(x)} H[p(y_g|x)], \quad (18)$$

$$SI(x) = SH(x) - SEH(x), \quad (19)$$

where the set of structure voxels $G(x)$ is now estimated from the posterior predictive distribution $p(y_{i,j,k} = GTVp|x)$, instead of the Monte Carlo samples of predictive distribution $p(y_{i,j,k} = GTVp|x, \theta)$.

**Uncertainty performance evaluation.** To evaluate the utility of the patient-level uncertainty, we performed multiple experiments described in the literature. First, similar to ref. 22, we developed a linear regression model with the cross-validation DSC values as the independent variables and uncertainty values as the dependent variables. In addition, a linear model with uncertainty value being the dependent variable and DSC being the independent variable was evaluated with results reported in Supplementary Results on Supplementary Table 3 and Supplementary Fig. 1. We then defined a threshold between uncertain and certain segmentations as the uncertainty value that the model predicted for 0.61 DSC. The threshold value was selected at 0.61 DSC since it represents the average interobserver variability for GTVp segmentation on PET/CT data as per previous literature[14]. The patient-level uncertainty estimates were then compared to the DSC values computed on the holdout dataset, by quantifying the four possible combinations, i.e., the model is uncertain, and the segmentation is inaccurate ($n_{i,u}$), the model is uncertain and the segmentation is accurate ($n_{a,u}$), the model is certain and the segmentation is accurate ($n_{a,c}$), and the model is certain and the segmentation is inaccurate ($n_{i,c}$). We then computed the following measures proposed in ref. 40: conditional probability that the segmentation is accurate given that the model is certain $p(accurate \mid certain)$ and that the segmentation is inaccurate given that the model is uncertain $p(inaccurate|uncertain)$, which are defined as:

$$p(accurate|certain) = \frac{n_{a,c}}{n_{a,c} + n_{i,c}}, \quad (20)$$

$$p(inaccurate|uncertain) = \frac{n_{i,u}}{n_{i,u} + n_{a,u}} \quad (21)$$

In addition, we report the overall performance with Accuracy vs. Uncertainty ($AU$) measure:

$$AU = p(accurate, certain) + p(inaccurate, uncertain)$$
$$= \frac{n_{a,c} + n_{i,u}}{n_{a,c} + n_{a,u} + n_{i,c} + n_{i,u}}, \quad (22)$$

which provides the probability of the outcome being segmentation is accurate given that the model is certain or segmentation is inaccurate given that the model is uncertain, similarly to the Patch Accuracy vs. Patch Uncertainty measure of ref. 40. In addition, we examined the linear relationship between the DSC performance and patient-level uncertainty estimates by quantifying the Pearson correlation coefficient between the DSC values and model certainty, defined as negative uncertainty, and denoted as $-EH$, $-H$, $-I$, $-R_{DSC}$, $-SEH$, $-SH$, $-SI$, $-CV$ for the eight uncertainty measures.

We also examined uncertainty-based referral simulation that is common in uncertainty-aware classification tasks[45,48,55]. In the batch referral process, each patient is assigned an uncertainty score using one of the uncertainty measures and the patients are sorted based on the score. Then, the patients are removed from the set, one at a time, beginning from the highest uncertainty score, and after each removal, i.e., simulated referral, the performance measures are computed on the remaining set of patients. This process simulates a scenario where the patients for which the model has high uncertainty are referred for an expert for manual verification and/or

correction, while expecting higher performance for the remaining patients. This process is repeated until 10% of the patients are remaining, as we observed that the performance measure estimates an increase in stochasticity with fewer patients. As a summary score for the combination of segmentation performance and uncertainty quantification through batch referral, we evaluate the area under the referral curve with Dice similarity coefficient (R-DSC AUC) by averaging the segmentation performance over multiple referral thresholds, in a similar manner as performed by Band et al.[45] with accuracy.

In addition, we evaluate the uncertainty-based referral with an instance-based process, where scans are referred according to a predetermined uncertainty threshold $\tau$ calculated with the training data. This analysis is more oriented for practice, as the cases are flagged for high uncertainty with a predetermined threshold value, instead of a value corresponding to a certain percentile computed on a batch of data, i.e., the holdout test set. We examine this process with three uncertainty thresholds: $\tau$ at 0.80, 0.85, and 0.90 cross-validation DSC, denoted as $\tau_{0.80}$, $\tau_{0.85}$, and $\tau_{0.90}$, respectively. The uncertainty thresholds are calculated for each uncertainty measure independently. As the instance referral process does not control the number of patients, but we would like for the model to confidently and accurately segment as many patients as possible, we also report the number of patients considered as certain in the instance referral process analysis.

## Results
### Segmentation performance
As for the overall segmentation performance without considering uncertainty, the MC Dropout Ensemble had the mean DSC value of 0.751 (SEM: 0.023) on the cross-validation data and 0.720 (IQR: 0.172) on the holdout data. The Deep Ensemble DSC was found to be 0.755 (SEM: 0.019) on the cross-validation data and 0.716 (IQR: 0.177) on the holdout data. As for the MSD metric, the MC Dropout Ensemble had the mean 2.01 mm (SEM: 0.295) on the cross-validation data and 2.31 (IQR: 1.15) mm on the holdout data, whereas the Deep Ensemble had the mean 2.07 mm (SEM: 0.271) on the validation data and 2.27 (IQR: 1.47) mm on the holdout data. For the MC Dropout Ensemble and the Deep Ensemble, the 95HD metric had the mean 6.49 mm (SEM: 1.14) and 6.89 mm (SEM: 1.11) on the cross-validation data, and 8.25 (IQR: 3.74) mm and 8.00 (IQR: 4.04) mm on the holdout data, respectively. There was a statistically significant difference ($P \le 0.05$) between the Deep Ensemble and the MC Dropout Ensemble for all the metrics on the holdout set with effect sizes of $-0.37$, $-0.30$, and $-0.18$ for DSC, MSD, and 95HD, respectively. Overall segmentation performance for the holdout data is illustrated in Fig. 2.

### Uncertainty estimation
In the case of the performance estimation with uncertainty, the MC Dropout Ensemble had highest $p(inaccurate \mid uncertain)$, $p(accurate \mid certain)$, and $AU$ values of 90.9% with $SEH$, 80.0% with $CV$, and 86.6% with $CV$, respectively. The Deep Ensemble had the highest value for $p(inaccurate \mid uncertain)$ with $SEH$, $p(accurate \mid certain)$ with $CV$, and $AU$ with $CV$, giving rise to values of 92.1%, 80.0%, and 86.6%, respectively. The $R_{DSC}$, $SH$, $SI$, and $I$ uncertainty measures had similar but slightly worse performance than the $CV$, on both the MC Dropout Ensemble and Deep Ensemble models. The worst performing uncertainty measure was **EH** with $AU$ values of 38.8% and 37.3% for MC Dropout Ensemble and Deep Ensemble, respectively. Overall for both models the $EH$, $H$, and $SEH$ uncertainty measures performed worse than other measures with values for $AU$ in the ranging from 37.3% to 65.6% while other measures ranged from 80.6% to 86.6%. In this analysis, the uncertainty thresholds were calculated at the 0.61 DSC on the cross-validation performance with values of $-1.7e-03$ for $-EH$, $-2.7e-03$ for $-H$, $-1.1e-03$ for $-I$, $-0.21$ for $-CV$, $-0.38$ for $-SEH$, $-0.49$ for $-SH$, $-0.10$ for $-SI$, and 0.74 for $R_{DSC}$ with MC Dropout Ensemble. The uncertainty thresholds for Deep Ensemble were $-1.4e-03$ for $-EH$, $-2.0e-03$ for $-H$, $-6.0e-04$ for $-I$, $-0.20$ for $-CV$, $-0.39$ for $-EH$, $-0.47$ for $-SH$, $-0.08$ for $-SI$, and 0.79 for $-R_{DSC}$ with the Deep Ensemble.

**Fig. 2 | Segmentation performance of the two approximate Bayesian methods.** Violinplot of Dice similarity coefficient (DSC), mean surface distance (MSD), and Hausdorff distance at 95% (95HD) performance on the external dataset (*N* = 67). The dotted lines mark the quartiles. Statistical significance is measured using the Wilcoxon signed-rank test.

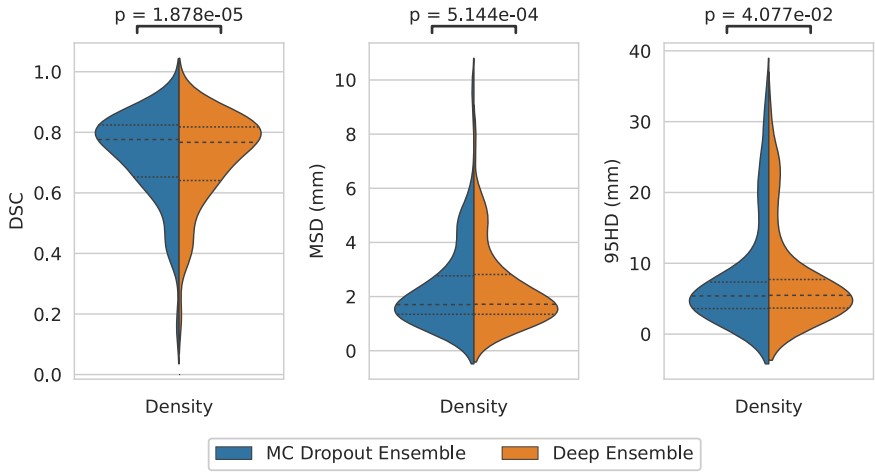

Full results are shown in Fig. 3 and Table 1. From the linear correlation analysis between DSC and certainty, *-CV* had a correlation of $\rho = 0.718$ ($P = 7.89\text{e-}12$) and $\rho = 0.720$ ($P = 6.64\text{e-}12$) for MC Dropout Ensemble and Deep Ensemble, respectively. Meanwhile, *-SEH* had a correlation value to the DSC with $\rho = -0.043$ ($P = 7.32\text{e-}01$) and $\rho = 0.465$ ($P = 7.24\text{e-}05$), for MC Dropout Ensemble and Deep Ensemble, respectively. The structure predictive entropy *-SH* had a correlation value to the DSC with $\rho = 0.704$ ($P = 2.99\text{e-}11$) and $\rho = 0.676$ ($P = 3.47\text{e-}10$), for MC Dropout Ensemble and Deep Ensemble, respectively. The structure mutual information *- SI* had a correlation value to the DSC with $\rho = 0.676$ ($P = 3.36\text{e-}11$) and $\rho = 0.623$ ($P = 1.84\text{e-}08$), for MC Dropout Ensemble and Deep Ensemble, respectively. Lastly, the DSC-risk *-$R_{DSC}$* had a correlation value of $\rho = 0.698$ ($P = 5.41\text{e-}11$) and $\rho = 0.704$ ($P = 3.13\text{e-}11$) for MC Dropout Ensemble and Deep Ensemble, respectively. Expected entropy *-EH* had a correlation value to the DSC with $\rho = -0.361$ ($P = 2.7\text{e-}03$) and $\rho = -0.388$ ($P = 1.2\text{e-}03$), for MC Dropout Ensemble and Deep Ensemble, respectively. The volume-level predictive entropy *-H* had a correlation value to the DSC with $\rho = -0.258$ ($P = 3.5\text{e-}02$) and $\rho = -0.303$ ($P = 1.3\text{e-}02$), for MC Dropout Ensemble and Deep Ensemble, respectively. The volume-level mutual information *- I* had a correlation value to the DSC with $\rho = 0.257$ ($P = 3.6\text{e-}02$) and $\rho = 0.303$ ($P = 1.3\text{e-}02$), for MC Dropout Ensemble and Deep Ensemble, respectively.

When simulating the batch referral process on the holdout set, by rejecting the most uncertain scans up to 90% of the total number of scans, it turned out that the coefficient of variation had the highest R-DSC AUC and expected entropy had the lowest R-DSC AUC for both of the models. During the referral process, all the uncertainty measures generally increased the performance, except for the structure expected entropy with MC Dropout Ensemble, as the DSC decreased under the initial, i.e., full holdout set, performance around 35% and past 85% referred cases. The batch referral curves for each uncertainty measure and the model are presented in Fig. 4. The R-DSC AUC performance was 0.681 with *EH*, 0.696 with *H*, 0.727 with *I*, 0.782 with *CV*, 0.752 with *SEH*, 0.771 with *SH*, 0.772 with *SI*, and 0.769 with *$R_{DSC}$* for MC Dropout Ensemble. For the Deep Ensemble, the R-DSC AUC performance was 0.672 with *EH*, 0.684 with *H*, and 0.727 with *I*, 0.783 with *CV*, 0.734 with *SEH*, 0.774 with *SH*, 0.778 with *SI*, and 0.775 with *$R_{DSC}$*.

For the instance-based referral process with the $\tau_{0.80}$ uncertainty threshold, the MC Dropout Ensemble referred a single patient with *H* and *I* measures that improved DSC to 0.728, while the Deep Ensemble did not refer any patients. For the $\tau_{0.85}$ uncertainty threshold, the MC Dropout Ensemble had the highest DSC value of 0.769 with the *CV* measure and 47 patients retained, while the Deep Ensemble had the highest DSC value of 0.827 with *I* measure and 3 patients retained. For the $\tau_{0.90}$ uncertainty threshold, the MC Dropout Ensemble had the highest DSC value of 0.876 with *CV* and 2 patients retained and for the Deep Ensemble the highest DSC value of 0.808 was obtained with *$R_{DSC}$* and 11 patients retained. Full results

for all uncertainty measures for both models and all thresholds are shown in Table 2.

In addition, we analyzed the uncertainty estimation in terms of the two distance-based measures, MSD and 95HD, that is reported in the Supplementary Methods, with Supplementary Table 4 and Supplementary Fig. 2 for MSD and Supplementary Table 5 and Supplementary Fig. 3 for 95HD. Overall, the best-performing uncertainty measures were the mutual information and coefficient of variation. However, all uncertainty measures performed similarly to these with the exception of the predictive entropy, expected entropy, and structure expected entropy that had the worst performance similarly to the main analysis using DSC.

When visually examining the voxel-wise uncertainty measures of predictive entropy, mutual information, and expected entropy for both MC Dropout Ensemble and Deep Ensemble models, the uncertainty is highest around the edges of the predicted segmentation mask for all the measures. Mutual information is mainly focused on the edges with the inner volume having high confidence, while expected entropy demonstrates moderate uncertainty near the inner volume. Full visual comparison of axial slices is shown in Fig. 5. Additional in-depth qualitative analysis of uncertainty maps for select cases are shown in Supplementary Results, Supplementary Figs. 4–7.

## Discussion

In this study, we have systematically investigated uncertainty-aware deep learning for OPC GTVp segmentation. We examined ways to leverage the uncertainty information in this segmentation task, specifically the uncertainty-based patient-level DSC performance estimation, utilization of uncertainty for automated referral for manual correction, and visualization of voxel-level uncertainty information. We implemented two established variations of Bayesian inference in DL, namely the MC Dropout Ensemble and the Deep Ensemble. Through experimentation with a multitude of uncertainty quantification measures (coefficient of variation, expected entropy, predictive entropy, mutual information, structure expected entropy, structure predictive entropy, structure mutual information, and DSC-risk) we compare and contrast differences between these approaches. Given the relative sparsity of existing literature for uncertainty estimation in OPC-related segmentation, our results act as an essential first benchmarking step towards a deeper understanding and further implementation of these techniques for clinically applicable radiotherapy segmentation workflows.

In terms of segmentation performance without considering uncertainty, both of the evaluated methods had similar performance while outperforming the expected average expert interobserver variability at 0.61 DSC. While the methods had statistical significant differences in performance, it should be noted that these minor differences are likely not clinically meaningful. Generally speaking, the state-of-the-art performance for the DL-based PET/CT OPC GTVp segmentation has remained mostly

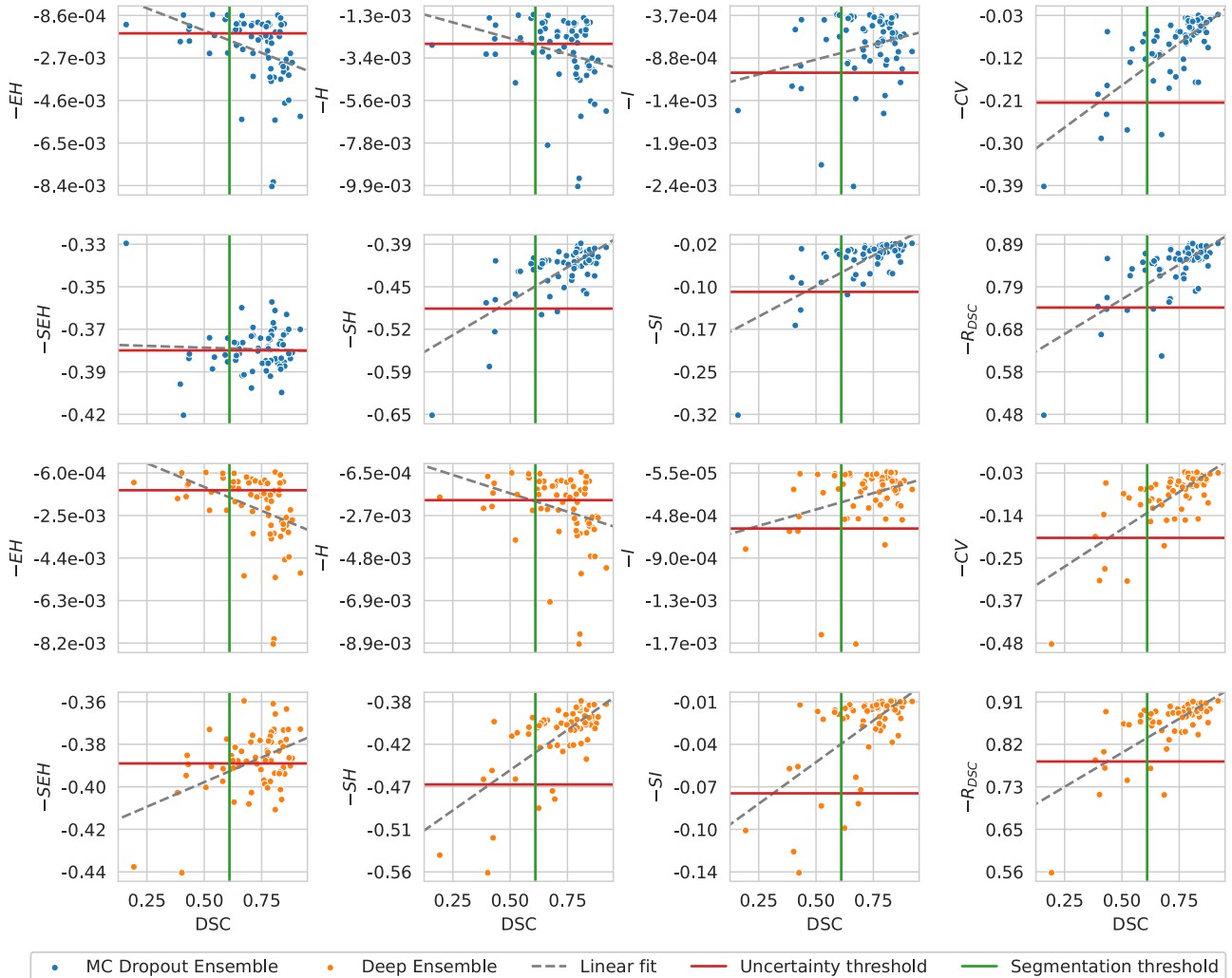

**Fig. 3 | Correlation analysis between model segmentation performance and model certainty.** Segmentation performance is based on Dice similarity coefficient (DSC) and model certainty (i.e., negative uncertainty) is based on expected entropy (EH), predictive entropy (H), mutual information (I), coefficient of variation (CV), structure expected entropy (SEH), structure predictive entropy (SH), structure mutual information (SI), or DSC-risk ($R_{DSC}$). The segmentation and uncertainty thresholds are drawn at the interobserver variability value of 0.61 DSC and at the predicted certainty value of 0.61 cross-validation DSC, respectively. Note that the x-axis is shared between the columns.

stagnant over the past few years, with external validation results being within a similar range as ours[51,54]. This is likely secondary to the already established large interobserver variability in OPC tumor-related segmentation. This further emphasizes the need for methods to provide clinicians with uncertainty estimates that could be used to further guide their clinical decision-making.

The uncertainty estimation performance between the two models was similar across most of the uncertainty measures, which suggests that the Deep Ensemble can be considered to be as accurate in the uncertainty estimation as the MC Dropout Ensemble, while requiring less computational resources. Indeed, our results are well aligned with a recent large-scale study of Bayesian DL, which showed that the Deep Ensemble is a competitive approximation for Bayesian inference[37]. As our uncertainty quantification methods require no modifications to the training of these models, and as the Deep Ensemble has been a popular approach for PET/CT OPC tumor segmentation (17 out of the 22 teams participating in the 2021 HECKTOR segmentation challenge utilized model ensembling[51]), it is straightforward to apply our uncertainty quantification framework for existing models to enable the model confidence to be used in practice.

Among all the uncertainty measures investigated, the coefficient of variation was generally favorable in terms of the performance estimation

with uncertainty and both of the referral processes, while the predictive entropy, expected entropy, and structure expected entropy had generally the worst performance. However, all the measures had a tendency to be over-confident in the uncertainty-based performance estimation, which is seen as a gap between the generalization and cross-validation DSC values. The poor expected entropy performance suggests that most of the uncertainty in this task is related to the model uncertainty, as the expected entropy has been described to capture the aleatoric component of uncertainty[45]. This is also suggested by the improved performance of the structure expected entropy with the Deep Ensemble in comparison to the MC Dropout Ensemble. The dropout mechanism of MC Dropout Ensemble covers the parameter space with much larger support than the samples of Deep Ensemble that could in turn identify more of the uncertainty as parameter-related, which is not captured by the structured expected entropy. When comparing the structure and standard entropy measures, all structure-based measures perform better than the standard counterparts. One probable cause for this can be the considerably higher class imbalance in 3D segmentation in comparison to 2D setting. Indeed, since the entropy can be seen as the expected negative log-likelihood of the predictive distribution, which is a well known surrogate loss for misclassification rate[60], it is expected that it is most influenced by the majority class. However, deeper analysis of these measures warrants a

**Table 1 | Conditional probabilities and overall accuracy for segmentation accuracy based on uncertainty**

| Model | MC Dropout Ensemble | | | Deep Ensemble | | |
|---|---|---|---|---|---|---|
| Uncertainty measure | p(accurate|certain) (%) | p(inaccurate|uncertain) (%) | AU (%) | p(accurate|certain) (%) | p(inaccurate|uncertain) (%) | AU (%) |
| EH | 75.0 | 12.8 | 38.8 | 74.1 | 12.5 | 37.3 |
| H | 80.0 | 15.6 | 49.3 | 77.1 | 12.5 | 46.3 |
| I | 86.2 | 44.4 | 80.6 | 86.9 | 66.7 | 85.1 |
| CV | 87.1 | **80.0** | **86.6** | 87.1 | **80.0** | **86.6** |
| SEH | **90.9** | 26.5 | 58.2 | **92.1** | 31.0 | 65.7 |
| SH | 85.5 | 60.0 | 83.6 | 85.2 | 50.0 | 82.1 |
| SI | 85.7 | 75.0 | 85.1 | 86.9 | 66.7 | 85.1 |
| $R_{DSC}$ | 86.9 | 66.7 | 85.1 | 86.9 | 66.7 | 85.1 |

Accurate/inaccurate is determined by the 0.61 DSC threshold, and certain/uncertain is determined by the predicted confidence threshold at 0.61 the cross-validation DSC. The overall segmentation accuracy is defined as the accuracy vs uncertainty (*AU*). Best results for each model are in bold.

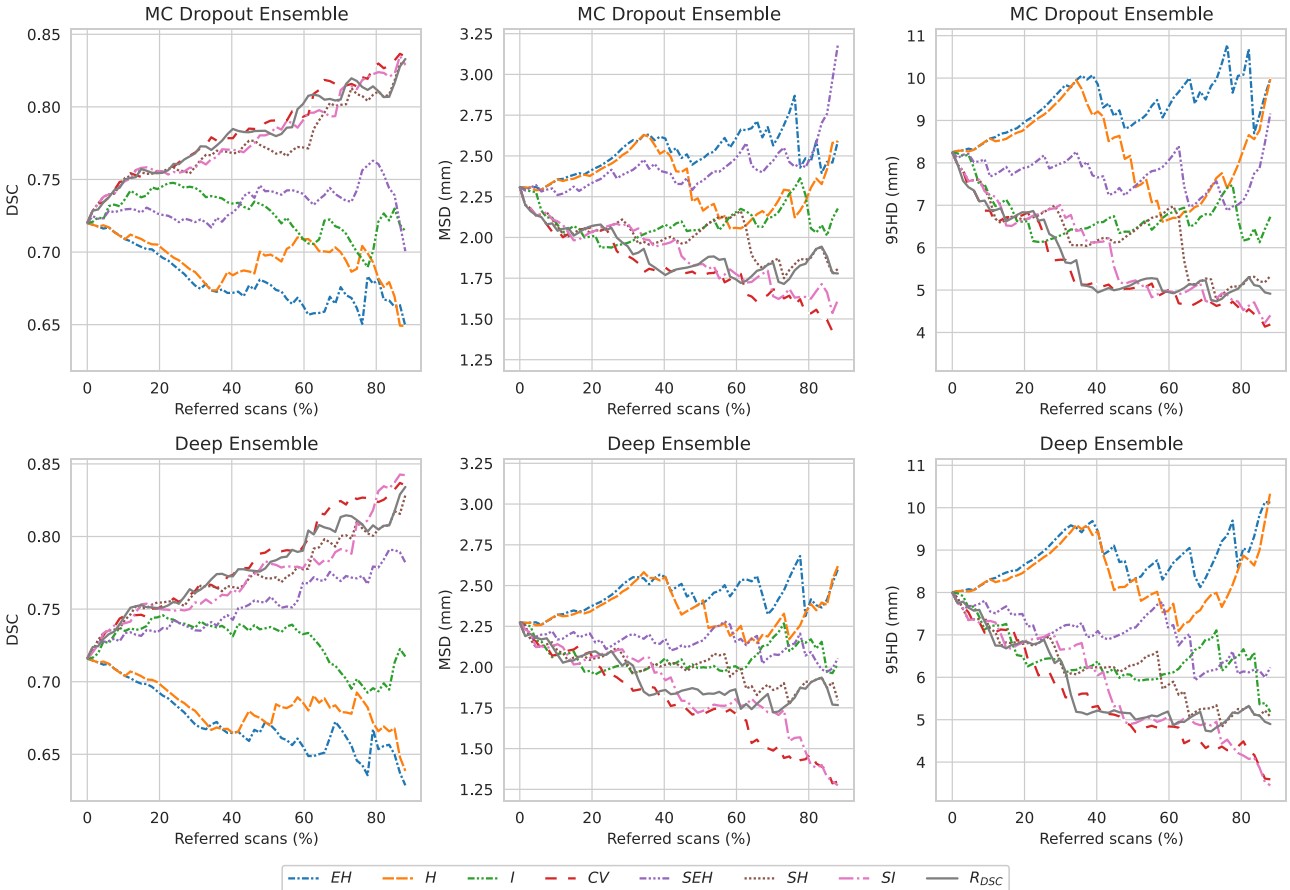

**Fig. 4 | Model segmentation performance in the batch referral process based on uncertainty measures.** Evaluated with Dice similarity coefficient (DSC), mean surface distance (MSD), and Hausdorff distance at 95% (95HD). Most uncertain scans are referred, based on expected entropy (*EH*), predictive entropy (*H*), mutual information (*I*), coefficient of variation (*CV*), structure expected entropy (*SEH*), structure predictive entropy (*SH*), structure mutual information (*SI*), and DSC-risk ($R_{DSC}$), up to 90% of the total scans.

more technical analysis that is out of scope of this study. The risk-based uncertainty measure DSC-risk had comparable performance to the structure uncertainty measures, but it was outperformed by the coefficient of variation in most of the experiments. The expected conditional risk used to develop the measure could be extended to reflect clinical preference or risk-averseness, such as the shape, size, and location of the model output segmentation, to fine-tune the uncertainty estimation for the specific OPC tumor segmentation task.

From our qualitative analysis of the uncertainty estimation, both of the methods produced uncertain voxels mainly on the edges of the predicted

segmentation mask for the expected entropy, i.e., the aleatoric component, mutual information, i.e., the epistemic component, and predictive entropy that includes both components of uncertainty. When comparing the models, the MC Dropout Ensemble method provided a smoother uncertainty gradient and more variation in the uncertainty, likely due to providing 300 samples per voxel compared to the five samples of Deep Ensemble. Moreover, a key takeaway from additional qualitative analysis includes a general overemphasis of PET signal by the models that could lead to erroneous predictions and uncertainty quantification. However, this finding is somewhat expected since studies of PET/CT auto-segmentation have

**Table 2 | Segmentation performance after threshold-based referral**

| Model | MC Dropout Ensemble | | | Deep Ensemble | | |
|---|---|---|---|---|---|---|
| Uncertainty measure/threshold | $\tau_{0.80}$ | $\tau_{0.85}$ | $\tau_{0.90}$ | $\tau_{0.80}$ | $\tau_{0.85}$ | $\tau_{0.90}$ |
| EH | 0.720 (N = 67) | — (N = 0) | — (N = 0) | **0.716** (N = 67) | — (N = 0) | — (N = 0) |
| H | 0.720 (N = 67) | — (N = 0) | — (N = 0) | **0.716** (N = 67) | — (N = 0) | — (N = 0) |
| I | 0.720 (N = 67) | — (N = 0) | — (N = 0) | **0.716** (N = 67) | **0.827** (N = 3) | — (N = 0) |
| CV | 0.720 (N = 67) | **0.769** (N = 47) | **0.876** (N = 2) | **0.716** (N = 67) | 0.752 (N = 53) | — (N = 0) |
| SEH | 0.720 (N = 67) | — (N = 0) | — (N = 0) | **0.716** (N = 67) | — (N = 0) | — (N = 0) |
| SH | **0.728** (N = 66) | 0.754 (N = 55) | 0.807 (N = 12) | **0.716** (N = 67) | 0.751 (N = 56) | 0.805 (N = 13) |
| SI | **0.728** (N = 66) | 0.755 (N = 51) | 0.835 (N = 9) | **0.716** (N = 67) | 0.750 (N = 48) | — (N = 0) |
| $R_{DSC}$ | 0.720 (N = 67) | 0.755 (N = 55) | 0.807 (N = 11) | **0.716** (N = 67) | 0.753 (N = 52) | **0.808** (N = 11) |

Segmentation performance based on Dice similarity coefficient (DSC) of remaining samples after thresholding with uncertainty. The uncertainty threshold ($\tau_{DSC}$) is based on cross-validation DSC at values of 0.80, 0.85, and 0.90. The number of patients retained after referral in parenthesis. The best results for each threshold and model are in bold.

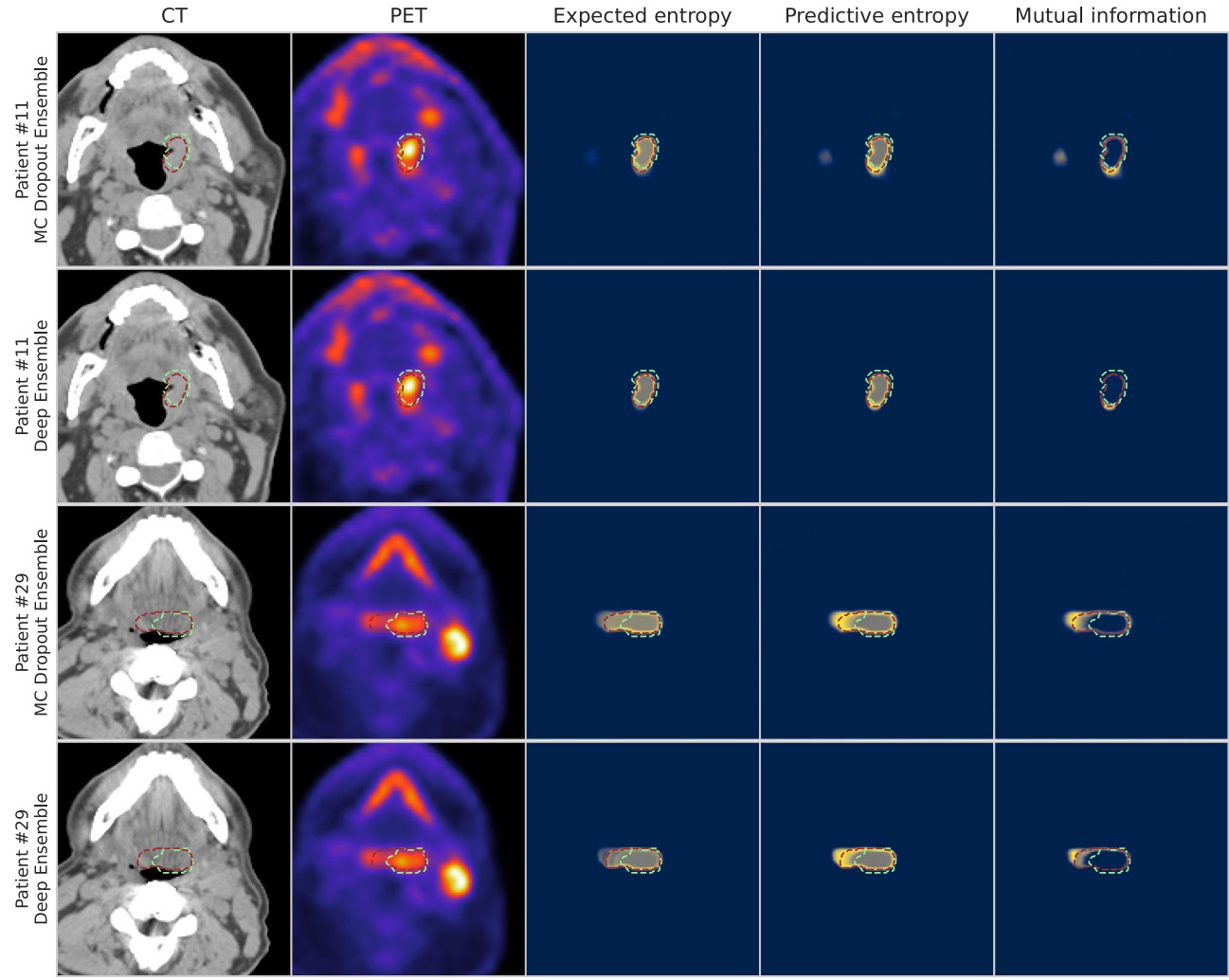

**Fig. 5 | Visualization of segmentations overlaid on top of the input modalities and uncertainty maps for two patients from the external validation set.** Patient #11 and Patient #29 correspond to a T1N2M0 and T2N2M0 tumor case, respectively; both patients had tumors that originated in the base of tongue region. The columns illustrate in order: computed tomography (CT), positron emission tomography (PET), expected entropy, predictive entropy, and mutual information. The model and expert segmentations are superimposed in red and green, respectively. Rows one and three contain the results for MC Dropout Ensemble while rows two and four contain the results for the Deep Ensemble. Blue, gray, and yellow colors in uncertainty maps correspond to low, medium, and high model uncertainty, respectively.

demonstrated that models generally utilize PET signal to a higher degree in model predictions compared to CT signal[61,62].

Lei et al.[33], Tang et al.[32], and van Rooij et al.[34] are among the few studies that have investigated segmentation-related uncertainties in HNSCC. Similar to our work, these studies utilized ensembling (Lei et al.[33]) and MC Dropout (Tang et al.[32], van Rooij et al.[34]) to segment nasopharyngeal cancer tumors and organs at risk on CT images, respectively. Moreover, in the only currently published study on the topic of uncertainty estimation in OPC GTVp segmentation, De Biase et al.[35] proposed a novel DL-based method using PET/CT images that generated probability maps for capturing the model uncertainty. The sequences of three consecutive 2-dimensional slices and the corresponding tumor segmentations were used as inputs to a model that leveraged inter/intra-slice context using attention mechanisms and recurrent neural network architectures. In their study, ensembling was used to derive probability maps rather than uncertainty maps, whereupon the authors experimented with different probability thresholds corresponding to areas of higher or lower agreement among the trained models. Our study acts as an important adjunct to the study by De Biase et al.[35], as the various methodologies investigated herein could be coupled with their proposed clinical solution.

There are some limitations in our study. First, although there are numerous probabilistic DL methods, we examined only two commonly used ones. These methods were selected due to their relative prevalence in existing literature and were thus deemed as an important starting point for exploring uncertainty estimation in OPC tumor-related segmentation. Second, we have utilized a relatively limited sample size for model training and evaluation. However, this study contains a robust training set from multiple institutions as supplied by the de-facto standard data science competition for OPC segmentation (i.e., HECKTOR) with external validation through our own institutional holdout dataset. Notably, our external validation dataset only contained human papillomavirus-positive patients; additional stratified analyses should be performed in the future studies using larger heterogenous external datasets. Moreover, we have chosen to utilize bounding boxes around the GTVp, as was performed for 2021 HECKTOR Challenge, in order to simplify the segmentation problem and focus on the exploration of uncertainty estimation whereas future studies should attempt the integration of uncertainty estimation into fully developed OPC segmentation workflows that can be applied to "as encountered" PET/CT images. Third, we have limited our investigation to the primary tumor, and not investigated nodal metastasis in this study, but as the newer editions of the HECKTOR Challenge includes these regions of interest, the incorporation of nodal metastasis should be the focus of future studies. Fourth, we have only investigated segmentations generated by a single observer for each scan, but the influence of multi-observer segmentations on uncertainty estimates is a future research direction. Finally, we have chosen to squarely focus on PET/CT as an imaging modality due to its ubiquity in OPC GTVp segmentation workflows. However, it is known that different imaging modalities (e.g., magnetic resonance imaging, contrast-enhanced CT) can provide complementary information for OPC tumor segmentation[63], and the combination of multiple image inputs may affect auto-segmentation model outputs[10,61,64]. Therefore, future research should investigate how results differ for models using alternative imaging modalities and the impacts of individual channel inputs on the uncertainty estimation.

## Conclusions

We applied Bayesian DL models with various uncertainty measures for OPC GTVp segmentation using multimodal large-scale datasets in order to evaluate the utility of uncertainty estimation. We found that regardless of the uncertainty measure applied, both of the models (Deep Ensemble and MC Dropout Ensemble) provided similar utility in terms of predicting segmentation quality and referral performance; due to its slightly lower computational cost and greater ubiquity, Deep Ensemble may be preferable to MC Dropout Ensemble. Notably, the coefficient of variation had overall favorable performance for both models so may be ideal as an uncertainty measure. While research in uncertainty estimation for OPC GTVp auto-segmentation is in its nascent stage, we anticipate that uncertainty

estimation will become increasingly important as these AI-based technologies begin to enter clinical workflows. Therefore our benchmarking study is a crucial first-step toward a wider adoption and exploration of these techniques. Future studies should investigate further uncertainty quantification methodology, larger sample sizes, additional relevant segmentation targets (i.e., metastatic lymph nodes), and incorporation of additional imaging modalities.

## Data availability

The HECKTOR 2021 training dataset[51] is publicly accessible from https://www.aicrowd.com/challenges/miccai-2021-hecktor requiring to fill in and sign an End User Agreement. The anonymized external validation dataset is publicly available on Figshare (https://doi.org/10.6084/m9.figshare.22718008) under CC BY 4.0 license[65]. CSV files used to create the Figures are available in Figshare (https://doi.org/10.6084/m9.figshare.22718008)[65].

## Code availability

All analyses were performed using the cited software, packages, and pipelines, whose codes were publicly available. The DL architecture in both models used the same 3D residual U-net from the Medical Open Network for AI (MONAI) (0.7.0)[52] (https://github.com/Project-MONAI/MONAI) Statistical comparisons and their annotations on figures were performed using the SciPy (1.7.3)[57] (https://github.com/scipy/scipy) and statannotations (0.4.4)[58] (https://github.com/trevismd/statannotations) Python packages, respectively.

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

## Acknowledgements

The work of Joel Jaskari, Jaakko Sahlsten, and Kimmo K. Kaski was supported in part by the Academy of Finland under Project 345449. Antti Mäkitie is supported in part by a grant from the Finnish Society of Sciences and Letters. Kareem A. Wahid is supported by the Dr. John J. Kopchick Fellowship through The University of Texas MD Anderson UTHealth Graduate School of Biomedical Sciences, the American Legion Auxiliary Fellowship in Cancer Research, and an NIH/National Institute for Dental and Craniofacial Research (NIDCR) F31 fellowship (1 F31DE031502-01) and the NCI NRSA Image Guided Cancer Therapy Training Program (T32CA261856). Benjamin H. Kann is supported by an NIH/National Institute for Dental and Craniofacial Research (NIDCR) K08 Grant (K08DE030216). Clifton D. Fuller receives related grant support from the NCI NRSA Image Guided Cancer Therapy Training Program (T32CA261856), as well as additional unrelated salary/effort support from NIH institutes. Dr. Fuller receives grant and infrastructure support from MD Anderson Cancer Center via: the Charles and Daneen Stiefel Center for Head and Neck Cancer Oropharyngeal Cancer Research Program; the Program in Image-guided Cancer Therapy; and the NIH/NCI Cancer Center Support Grant (CCSG) Radiation Oncology and Cancer Imaging Program (P30CA016672). Dr. Fuller has received unrelated direct industry grant/in-kind support, honoraria, and travel funding from Elekta AB.

## Author contributions

Study concepts: all authors; study design: J.S., K.W., M.N., and J.J.; data acquisition: S.A., K.W., M.N., and R.H.; quality control of data and algorithms: J.S. and E.G.; data analysis and interpretation: J.S., K.W., J.J., B.K., A.M., and K.K.; manuscript editing: J.S., J.J., K.W., M.N., E.G., B.K., A.M., K.K., and C.F. All authors contributed to the article and approved the submitted version.

## Competing interests

The authors declare no competing interests.
