## [Peer Review File · Communications Medicine]

Reviewers' comments:

Reviewer #1 (Remarks to the Author):

The authors examine the use of DL uncertainty estimation for primary gross tumor volume (GTVp) segmentation in oropharyngeal cancer (OPC). They use two models (Deep Ensemble and Monte Carlo Drop Out), five uncertainty measures (coefficient of variation, mean pairwise Dice, structured expected entropy, predictive entropy, and mutual information), and two publicly available datasets of PET/CT scan.

While their proposal is not novel, uncertainty management is a very interesting topic that has not been widely applied to this task and their work might be of interest to researchers focusing on this problem. As they remark in the paper, there might be a gap in knowledge at OPC GTVp segmentation. However, I have some concerns about their work, such as the completeness of the paper and the clarity of some ideas. I think that the goal of the paper has not been clearly defined which raises the following issues:

- If the paper is intended to bring uncertainty estimation to unfamiliar readers, it needs to explain in more detail some basic ideas.

- If the paper is intended to present a new approach for OPC GTVp segmentation, the model should be explained in detail and it should be compared with other approaches in the state of the art.

- If the paper is intended to propose a new uncertainty measure, more experiments and justification are needed, as the results show that the proposed measure has comparable performance to most established measures and it is outperformed by the coefficient of variation.

Specific comments, with recommendations for addressing each comment

The contributions of this work are not clear. According to the abstract and conclusions, the authors have applied DL models in order to evaluate the utility of uncertainty estimation and make it visible to clinicians and researchers in this area. However, some key ideas such as aleatoric/epistemic uncertainty and the basis of Bayesian DL approaches (Deep Ensemble and Monte Carlo Dropout) are not properly included. Including more detailed information would be of interest to the unfamiliar reader (if this is the target audience).

The abstract mentions an “accurate prediction of the quality of DL segmentation in 86.6% of cases. This data never appears again in the paper and it is not justified.

The final model used is unclear. The last paragraph of “Bayesian DL Models” lines 180-186 states that hyperparameters were considered on the loss function, which is not included in the paper. The loss function needs to be explicitly included along with those parameters and their range.

Section “Uncertainty measures” require to be thoroughly revised:

Information theoretic entropy is presented as a widely used measure, then discarded. Even if there are other options, it is interesting to include this one in the study for easier comparison with other models or works.

The abstract introduces the idea that the authors will use five measures, but this section makes it confusing. I would appreciate an overview paragraph at the beginning. Instead, the authors first

present one measure to be discarded, then three (following 44,21), with their respective definitions (Ucv, Up, Umi), then a new one (Udr). One has to read again to find the missing one, the expected entropy, which is defined as Ue in Figure 3 (Next section).

Some metrics are defined in text and others as equations. The authors should follow unified criteria and include an explicit equation for each measure.

If Dice-risk uncertainty measure is a novel contribution of this paper. Why it is not mentioned before?

$p_{i,k,j}$ is defined in line 216, but then $p_{i,k,j}(m)$ is used in l.220. $p_{i,k,j}(m)$ needs to be explicitly defined. The kind of uncertainty (total, aleatoric, epistemic) for which each measure accounts, should be moved to the end of the section, clearly stating the goal of each of the five measures to be used. The acronym DSC is used here, but defined in the next section.

Line 247 states that the authors use the Dice-loss as the loss function. Is this related to the training of the networks? It seems that it is just used as a measure and not as an actual loss function. the authors state that only a limited number of studies have investigated 3D uncertainty estimation. Would it be of interest to add further details on how 3D is addressed in this paper, and why?

The notation has to be revised. In some cases the probability of A given B is notated as $p(A \vee B)$ and in others $p(A|B)$. It should be consistent.

Paragraph 273-286 requires revision and further clarification. What is exactly the uncertainty value that the model predicted for 0.61 DSC? How does this evaluate the utility of uncertainty vs not using uncertainty? This question is raised here but not answered here nor in the discussion.

Similarly, in order to evaluate the utility of uncertainty, the authors should include the basic approach without uncertainty estimation in the "Segmentation Performance" experiment. What is the advantage in terms of segmentation of using uncertainty estimation?

"Performance evaluation", "Results", and "Discussion" would benefit from subsections that make it easier for the reader to distinguish between the different experiments. The separation is only used in the results sections which makes it more confusing.

Related to comment 3: After reading the results section, I wonder if the uncertainty measures are used during training and how. They seem to affect the model performance. This needs to be further clarified.

The proposed Dice-Risk uncertainty measure seems to be equivalent to other measures in the study. Why is it interesting? How does this metric contribute to the measure of uncertainty?

Lines 465-466 state a benefit of the proposed method in terms of processing time. This statement needs to be properly justified. How was this measured and against what?

Some statements should be accompanied by a proper citation. Some examples, but not all of them:
lines 80-81: which approaches demonstrate that?
lines 155-156: which studies make these approaches popular for medical segmentation?

Reviewer #2 (Remarks to the Author):

In the manuscript the authors investigated uncertainty-aware deep learning for oropharyngeal cancer (OPC) primary tumor segmentation in PET/CT images. They used a multi-center dataset of 224 OPC patients as training/validation set and a single-center dataset of 67 OPC as test set. In the study they:

1. compared the segmentation performance of two methods for Bayesian inference in DL, namely Deep Ensemble and MC drop out Ensemble;
2. quantitatively (using several metrics for uncertainty quantification) and qualitatively (visualizing voxel-wise uncertainty) assessed whether patient-level segmentation performance correlates with uncertainty quantification for both segmentation methods;
3. simulated an uncertainty-based referral strategy and compared the two segmentation methods according to the number of cases identified for referral by the uncertainty quantification metrics.

Overall, I find that the manuscript offers a comprehensive and thorough analysis. However, I provide some comments below with the intention to improve the manuscript further.

1. Introduction. Line 106: is it true that the approach realizes accurate OPC GTVp auto-segmentation? Looking at Figure 1, the last step requires the human intervention which makes the method not fully automatic. I would not mention auto-segmentation.
2. Method. As general comment, I found the method section, at times, hard to follow. A lot of implementation details were written and listed in the text. A lot of info from line 158 to line 168 could be summarized in a table where the two chosen methods can be compared.
3. Uncertainty measures and Performance evaluation are now two separate sections. However, the metrics used to quantify uncertainty are used as evaluation metrics too. I would either create one single section with different subsections (metrics to evaluate segmentation, to quantify uncertainty, etc..) or having line 260-271 before line 188 as a separate section. The metrics used to assess segmentation performance should be mentioned first, as in the Results section.
4. Line 134-135 what do you mean by one annotator per scan? Were all the scans annotated by all different clinicians?
5. From line 180-186 is not clear which hyperparameters were chosen at the end. This makes the experiments hard to reproduce.
6. Line 202: It should be ":" instead of ";" in "In the case of MC Dropout, Roy et al. 44 and Hoebel et al. 21, proposed three so-called structure-wise uncertainty measures: the coefficient of variation (CV), the mean pairwise Dice, and the structure expected entropy."
7. For U_E , U_P and U_{MI} there are not explicit formulas. In Line 215 should that be U_E ? Be consistent with your nomenclature.
8. Line 222-223, it says "over the voxels that were considered as positive based on the average of the Monte Carlo samples", what about for the deep ensembling? Is it different there?
9. Line 260-262: what do you mean by "specifically, the mean value of these metrics." after mentioning "mean surface distance" and "mean Hausdorff distance"?

10. Line 275: by HECKTOR DSC values you mean the “validation DSC values”? I would not write HECKTOR values, it is misleading
11. Uncertainty estimation: The uncertain/certain cases were compared to DSC only (Line 342). We know that DSC is not always the best metric to assess segmentation performance. Why did you choose to use DSC only? Did you try to use MSD and 95th HD also? How would your conclusion/results change if so? I would at least add something in the discussion about it.
12. Figure 3 is very small and hard to read. The legend is too small, I suggest to make the image bigger.
13. Line 390: I struggle to see the added value of having Table 2. The referral strategy is interesting but it is not mentioned at all in the discussion. How should we interpret Table 2 based on the results we obtained from Table 1 and Figure 3? I would spend some words on this in the discussion.
14. Expected entropy results obtained with deep ensemble and MC dropout ensemble are very different in Figure 3, besides being the worst compared to the others. Do you have an explanation for that?
15. Line 514: “De Biase et al.”
16. Line 518-519: I don’t see how using 5 metrics is a limitation.

Specific Reviewer Comments and Author's reply

We would like to thank the editors and reviewers of Communications Medicine for thoroughly reading our manuscript and providing valuable suggestions. We believe to have addressed the reviewers' concerns, and we have now improved the manuscript accordingly. Reviewers' comments are listed below, with our corresponding responses annotated in blue. All corrections and additions are highlighted in the edited manuscript file in yellow, green, and cyan corresponding to responses to Reviewer #1, Reviewer #2, and both reviewers, respectively.

Reviewer #1:

The authors examine the use of DL uncertainty estimation for primary gross tumor volume (GTVp) segmentation in oropharyngeal cancer (OPC). They use two models (Deep Ensemble and Monte Carlo Drop Out), five uncertainty measures (coefficient of variation, mean pairwise Dice, structured expected entropy, predictive entropy, and mutual information), and two publicly available datasets of PET/CT scan. While their proposal is not novel, uncertainty management is a very interesting topic that has not been widely applied to this task and their work might be of interest to researchers focusing on this problem. As they remark in the paper, there might be a gap in knowledge at OPC GTVp segmentation. However, I have some concerns about their work, such as the completeness of the paper and the clarity of some ideas. **Response:** We thank the Reviewer for the encouraging comments and constructive feedback that have considerably improved our manuscript.

I think that the goal of the paper has not been clearly defined which raises the following issues:

-If the paper is intended to bring uncertainty estimation to unfamiliar readers, it needs to explain in more detail some basic ideas.

-If the paper is intended to present a new approach for OPC GTVp segmentation, the model should be explained in detail and it should be compared with other approaches in the state of the art.

-If the paper is intended to propose a new uncertainty measure, more experiments and justification are needed, as the results show that the proposed measure has comparable performance to most established measures and it is outperformed by the coefficient of variation.

Response: We thank the Reviewer for summarizing the criticism of the main goal of the manuscript. We have now amended our manuscript to clarify our main aim which is to introduce the uncertainty in deep learning for a more broad audience. These include adding a new subsection "Uncertainty Estimation in Deep Learning" in the Introduction section, clarification of the "Uncertainty estimation" subsection in the Methods section, and adding a more fitting discussion.

Specific comments, with recommendations for addressing each comment

The contributions of this work are not clear. According to the abstract and conclusions, the authors have applied DL models in order to evaluate the utility of uncertainty estimation and make it visible to clinicians and researchers in this area. However, some key ideas such as aleatoric/epistemic uncertainty and the basis of Bayesian DL approaches (Deep Ensemble and Monte Carlo Dropout) are not properly included. Including more detailed information would be of interest to the unfamiliar reader (if this is the target audience).

Response: We thank the Reviewer for pointing out this missing information. We have now added a subsection named "Uncertainty Estimation in Deep Learning" in the Introduction section, which will introduce these important ideas in detail. We believe this has strengthened our manuscript for unfamiliar readers.

The abstract mentions an “accurate prediction of the quality of DL segmentation in 86.6% of cases. This data never appears again in the paper and it is not justified.

Response: We thank the Reviewer for the comment. This result was reported as a decimal value in Table 1 as the best value for Accuracy vs. Uncertainty (originally AvU, now AU) measure. We have now unified the number formatting throughout the manuscript to match the Abstract.

The final model used is unclear. The last paragraph of “Bayesian DL Models” lines 180-186 states that hyperparameters were considered on the loss function, which is not included in the paper. The loss function needs to be explicitly included along with those parameters and their range.

Response: We thank the Reviewer for pointing out this missing information. The hyperparameters of the final model are now included in the revised manuscript. In addition, the loss functions are now defined.

Section “Uncertainty measures” require to be thoroughly revised:

Information theoretic entropy is presented as a widely used measure, then discarded. Even if there are other options, it is interesting to include this one in the study for easier comparison with other models or works.

Response: We thank the Reviewer for this important comment. We have now revised and condensed the “Uncertainty measures” subsection. The information theoretic entropy measures are introduced in the new Introduction’s subsection “Uncertainty Estimation in Deep Learning”, while the “Uncertainty measures” in the Methods section focuses on more recent applications and refinements for the standard uncertainty measures used in this work. In addition, we have included results for the information theoretic entropy in the main manuscript. Moreover, we improved the structure of Table 2 (originally Table 3) due to additional uncertainty evaluation measures.

The abstract introduces the idea that the authors will use five measures, but this section makes it confusing. I would appreciate an overview paragraph at the beginning. Instead, the authors first present one measure to be discarded, then three (following 44,21), with their respective definitions (U_{cv} , U_p , U_{mi}), then a new one (U_{dr}). One has to read again to find the missing one, the expected entropy, which is defined as U_e in Figure 3 (Next section).

Response: We thank the Reviewer for this comment. We believe the new overview subsection “Uncertainty Estimation in Deep Learning” and the new version of subsection “Uncertainty measures” have clarified all of the evaluated uncertainty measures. In addition, we have now revised all notations to follow previous literature for predictive entropy (H), mutual information (I), and coefficient of variation (CV). Moreover, we used EH as an abbreviation for the expected entropy and R_{DSC} for the risk based measure DSC-risk. The structure variants include structure predictive entropy (SH), structure mutual information (SI), and structure expected entropy (SEH).

Some metrics are defined in text and others as equations. The authors should follow unified criteria and include an explicit equation for each measure.

Response: We thank the Reviewer for this comment. We have now amended the manuscript to include all used formulas and their derivations in explicit equations.

If Dice-risk uncertainty measure is a novel contribution of this paper. Why it is not mentioned before? $\pi_{i,k,j}$ is defined in line 216, but then $\pi_{i,k,j}(m)$ is used in l.220. $\pi_{i,k,j}(m)$ needs to be explicitly defined. The kind of uncertainty (total, aleatoric, epistemic) for which each measure accounts, should be moved to the end of the section, clearly stating the goal of each of the five measures to be used.

The acronym DSC is used here, but defined in the next section.

Response: We thank the Reviewer for this comment. We have now clarified our novel contributions in the Introduction section and added an introductory paragraph for the risk based measures in the Introduction section and definition for the DSC-risk measure in the Methods section. Moreover, we have now added information about each uncertainty type.

Line 247 states that the authors use the Dice-loss as the loss function. Is this related to the training of the networks? It seems that it is just used as a measure and not as an actual loss function.

the authors state that only a limited number of studies have investigated 3D uncertainty estimation. Would it be of interest to add further details on how 3D is addressed in this paper, and why?

Response: We thank the Reviewer for pointing out this ambiguity and comments. We have now distinctly written the loss functions, including Dice loss, which were used in model development and the evaluation measures in separate subsections. Moreover, we highlight that the Dice loss is a soft approximation to the DSC (Dice similarity coefficient) that separates these two functions from each other. We now use the DSC naming exclusively for the evaluation measure and Dice loss for the loss function. In addition, we have now added discussion about the considerations between 2D and 3D cases for uncertainty estimation in segmentation tasks.

The notation has to be revised. In some cases the probability of A given B is notated as $p(A \vee B)$ and in others $p(A|B)$. It should be consistent.

Response: We thank the Reviewer for this misleading notation. We have now clarified the definitions for the conditional probabilities using $p(A|B)$ nomenclature, and changed the Accuracy vs. Uncertainty abbreviation from AvU to AU to avoid confusion. In addition, the order of definition has been changed to reflect this change.

Paragraph 273-286 requires revision and further clarification. What is exactly the uncertainty value that the model predicted for 0.61 DSC? How does this evaluate the utility of uncertainty vs not using uncertainty? This question is raised here but not answered here nor in the discussion.

Response: We thank the Reviewer for pointing out this missing information and lack of clarity. We have now added values for the uncertainty measures at the 0.61 DSC threshold. This threshold was selected as it is the expected interobserver variability (as derived from existing literature), which means that if the uncertainty can be used to identify the performance for a certain scan as higher than 0.61 DSC, the model would perform better than the expert on average. Additionally, in the converse case, when the uncertainty information predicts low accuracy (less than 0.61 DSC), an expert could intervene and manually annotate the case with expected improvement on DSC. This type of analysis benefits considerably with well calibrated uncertainty-aware methods which is now clarified and discussed in the manuscript.

Similarly, in order to evaluate the utility of uncertainty, the authors should include the basic approach without uncertainty estimation in the “Segmentation Performance” experiment. What is the advantage in terms of segmentation of using uncertainty estimation?

Response: The experiments in the “Segmentation Performance” subsection do not utilize uncertainty information, which is now clarified in the main manuscript. Instead the results are for the full external dataset to provide the reader with baseline performances for the models. To observe the advantage of using uncertainty, we can select for example the Deep Ensemble that has 0.716 DSC for the full set, presented in the Segmentation Performance section, and compare this result to e.g., Figure 4, where we can see that when we use the uncertainty information of

CV , SI , SH , or R_{DSC} to identify 20% of the most uncertain cases, we can expect roughly 0.750 DSC for the remaining 80% of cases. This type of mechanism that identifies the poor performance cases can be utilized for example in quality control for initial auto-segmentation methods.

“Performance evaluation”, “Results”, and “Discussion” would benefit from subsections that make it easier for the reader to distinguish between the different experiments. The separation is only used in the results sections which makes it more confusing.

Response: We thank the Reviewer for pointing out this improvement for the manuscript structure. We have now restructured the Methods section with distinct subsections to clarify the different experiments. The new additions to the Discussion section have clarified the section sufficiently.

Related to comment 3: After reading the results section, I wonder if the uncertainty measures are used during training and how. They seem to affect the model performance. This needs to be further clarified.

Response: We thank the Reviewer for this insightful comment. We have now clarified in the manuscript that the uncertainty information was only used in model selection part of the model development and training by selecting the highest performing model based on the area of the DSC referral curve, a measure that combines the overall segmentation performance of the dataset and the level of improvement with uncertainty based referral. This choice was made to keep the analysis as general as possible.

The proposed Dice-Risk uncertainty measure seems to be equivalent to other measures in the study. Why is it interesting? How does this metric contribute to the measure of uncertainty?

Response: We thank the Reviewer for this comment. We have now clarified that there are currently three ways of thinking of uncertainty: the information theoretic approach of entropy, the risk -based approach, and the *ad hoc* design of functions that might benefit in the task. The entropy and risk -based approaches have clear theoretic justification: the entropy is a measure of “surprise” of a probability distribution, whereas the risk is a measure of “regret” associated with some cost of making an incorrect decision. There has been multiple recent works in deep learning uncertainty estimation using the risk -based approach, and thus we decided to include this approach to our analysis as well.

Lines 465-466 state a benefit of the proposed method in terms of processing time. This statement needs to be properly justified. How was this measured and against what?

Response: We thank the Reviewer for pointing out this incomplete description of the processing time analysis. As the analysis and results were not part of our main manuscript and only mentioned in the Discussion section, we have now moved this analysis to Appendix B, where it is explained in full.

Some statements should be accompanied by a proper citation. Some examples, but not all of them:

lines 80-81: which approaches demonstrate that?

lines 155-156: which studies make these approaches popular for medical segmentation?

Response: We have added literature evidence to our claims in the Introduction and Methods sections. Specifically, the mentioned lines have now been amended as follows:

lines 80-81:

OLD: *However, while there exists a deluge of DL-based OPC auto-segmentation approaches that demonstrate potentially clinically acceptable performance in terms of geometric measures,*

NEW: *However, while there exists a deluge of DL-based OPC auto-segmentation approaches that demonstrate potentially clinically acceptable performance in terms of geometric measures as evident from HECKTOR* ¹⁶

lines 155-156:

OLD: *Two common posterior approximation methods for segmentation in the literature are deep ensembling and Monte Carlo (MC) Dropout.*

NEW: *Two common posterior approximation methods for segmentation in the literature are deep ensembling* ^{22,38} *and Monte Carlo (MC) Dropout* ^{26,39,40}.

Added references:

16. Andrearczyk, V. *et al.* Automatic Head and Neck Tumor segmentation and outcome prediction relying on FDG-PET/CT images: Findings from the second edition of the HECKTOR challenge. *Med. Image Anal.* **90**, 102972 (2023).
22. Hoebel, K. *et al.* An exploration of uncertainty information for segmentation quality assessment. in *Medical Imaging 2020: Image Processing* vol. 11313 381–390 (SPIE, 2020).
26. Roy, A. G., Conjeti, S., Navab, N. & Wachinger, C. Inherent Brain Segmentation Quality Control from Fully ConvNet Monte Carlo Sampling. in *Medical Image Computing and Computer Assisted Intervention – MICCAI 2018* 664–672 (Springer International Publishing, 2018).
38. Mehrtash, A., Wells, W. M., Tempany, C. M., Abolmaesumi, P. & Kapur, T. Confidence Calibration and Predictive Uncertainty Estimation for Deep Medical Image Segmentation. *IEEE Trans. Med. Imaging* **39**, 3868–3878 (2020).
39. Hoebel, K., Chang, K., Patel, J., Singh, P. & Kalpathy-Cramer, J. Give me (un)certainty -- An exploration of parameters that affect segmentation uncertainty. *arXiv [eess.IV]* (2019).
40. Mukhoti, J. & Gal, Y. Evaluating Bayesian Deep Learning Methods for Semantic Segmentation. *arXiv [cs.CV]* (2018).

Reviewer #2:

In the manuscript the authors investigated uncertainty-aware deep learning for oropharyngeal cancer (OPC) primary tumor segmentation in PET/CT images. They used a multi-center dataset of 224 OPC patients as training/validation set and a single-center dataset of 67 OPC as test set. In the study they:

- 1.compared the segmentation performance of two methods for Bayesian inference in DL, namely Deep Ensemble and MC drop out Ensemble;
- 2.quantitatively (using several metrics for uncertainty quantification) and qualitatively (visualizing voxel-wise uncertainty) assessed whether patient-level segmentation performance correlates with uncertainty quantification for both segmentation methods;
- 3.simulated an uncertainty-based referral strategy and compared the two segmentation methods according to the number of cases identified for referral by the uncertainty quantification metrics.

Overall, I find that the manuscript offers a comprehensive and thorough analysis. However, I provide some comments below with the intention to improve the manuscript further.

Response: We thank the Reviewer for the supportive comments and precise feedback that has been used to improve the manuscript.

1.Introduction. Line 106: is it true that the approach realizes accurate OPC GTVp auto-segmentation? Looking at Figure 1, the last step requires the human intervention which makes the method not fully automatic. I would not mention auto-segmentation.

Response: We thank the Reviewer for this comment. The proposed uncertainty-aware deep learning neural network does provide OPC GTVp auto-segmentation with higher than interobserver variability performance on average. In contrast, the proposed uncertainty-based framework in Figure 1, uses the neural network to provide auto-segmentation results which can be improved by manual or automatic intervention. We have now clarified this distinction in the Introduction section.

2.Method. As general comment, I found the method section, at times, hard to follow. A lot of implementation details were written and listed in the text. A lot of info from line 158 to line 168 could be summarized in a table where the two chosen methods can be compared.

Response: We thank the Reviewer for pointing out the lacking coherence in our text. We acknowledge that this section describing the method can be difficult to follow due to the goal of concise reporting. We omitted the use of a comparison table as the methods have exactly the same architecture. However, we have now improved the readability of this subsection and clarified that the two methods use exactly the same architecture and only differ how they are used during inference.

3.Uncertainty measures and Performance evaluation are now two separate sections. However, the metrics used to quantify uncertainty are used as evaluation metrics too. I would either create one single section with different subsections (metrics to evaluate segmentation, to quantify uncertainty, etc..) or having line 260-271 before line 188 as a separate section. The metrics used to assess segmentation performance should be mentioned first, as in the Results section.

Response: We thank the Reviewer for pointing out this inconsistency in the manuscript structure. We have now divided the Methods section to include two separate subsections i.e., one for segmentation performance and one for uncertainty estimation, similar to the Results section.

4.Line 134-135 what do you mean by one annotator per scan? Were all the scans annotated by all different clinicians?

Response: We thank the Reviewer for pointing out this ambiguous description. We have now clarified that there were multiple annotators but each OPC GTVp was annotated by a single annotator.

5.From line 180-186 is not clear which hyperparameters were chosen at the end. This makes the experiments hard to reproduce.

Response: We thank the Reviewer for pointing out this missing information for reproducibility. We have now added the hyperparameters of the final model after defining the ranges on the Method section.

6.Line 202: It should be “:” instead of “;” in “In the case of MC Dropout, Roy et al. 44 and Hoebel et al. 21,proposed three so-called structure-wise uncertainty measures: the coefficient of variation (CV), the mean pairwise Dice, and the structure expected entropy.”

Response: We thank the Reviewer for pointing out the grammatical errors. We have now amended the errors and improved the overall language of our manuscript.

7.For U_E , U_P and U_{MI} there are not explicit formulas. In Line 215 should that be U_E ? Be consistent with your nomenclature.

Response: We thank the Reviewer for pointing out the inconsistent nomenclature. We have now amended the manuscript to include consistent formatting with explicit equations defining the uncertainty measures used throughout the manuscript. We have now revised all notations to follow previous literature for predictive entropy (H), mutual information (I), and coefficient of variation (CV). In addition, we used EH as an abbreviation for the expected entropy and R_{DSC} for the risk based measure DSC-risk. The structure variants include the structure predictive entropy (SH), structure mutual information (SI), and structure expected entropy (SEH).

8.Line 222-223, it says “over the voxels that were considered as positive based on the average of the Monte Carlo samples”, what about for the deep ensembling? Is it different there?

Response: In the Deep Ensemble approach, each ensemble member is thought to be a Monte Carlo sample from the modes of the posterior. This matter is now clarified in the manuscript.

9.Line 260-262: what do you mean by “specifically, the mean value of these metrics.” after mentioning “mean surface distance” and “mean Hausdorff distance”?

Response: We thank the Reviewer for pointing out this unclear text addition. The aim was to describe that all comparisons are evaluated in the dataset mean instead of e.g., dataset median. We now report the aggregation method in the Results section, thus this text is now removed from the Methods section to improve clarity.

10.Line 275: by HECKTOR DSC values you mean the “validation DSC values”? I would not write HECKTOR values, it is misleading

Response: We thank the Reviewer for pointing out this misleading wording. Original intention was to highlight the distribution shift between the dataset used for the cross-validation and the dataset used for external validation as they come from different hospitals. We have now revised the manuscript and fixed the reference to the results using the HECKTOR dataset as the cross-validation results.

11.Uncertainty estimation: The uncertain/certain cases were compared to DSC only (Line342). We know that DSC is not always the best metric to assess segmentation performance. Why did you choose to use DSC only? Did you try to use MSD and 95th HD also? How would your conclusion/results change if so? I would at least add something in the discussion about it.

Response: We thank the Reviewer for pointing out this limitation. We selected DSC as the main measure for this analysis to provide reasonable scope and length of the manuscript. We have now acknowledged this as a limitation in the manuscript and have now added uncertainty estimation with the MSD and 95HD as supplementary results in the Appendix B. With these distance based measures, the best performing uncertainty measure changes from coefficient of variation to mutual information or structure mutual information. However, this does not change the conclusions as all of the best performing uncertainty measures (I , CV , SH , SI , and R_{DSC}) perform within a small range to each other for both models, similarly to the main analysis with DSC.

12.Figure 3 is very small and hard to read. The legenda is too small, I suggest to make the image bigger.

Response: We thank the Reviewer for pointing out the small font sizes in the Figure. The Figure legend and text font sizes have now been increased.

13.Line 390: I struggle to see the added value of having Table 2. The referral strategy is interesting but it is not mentioned at all in the discussion. How should we interpret Table 2 based on the results we obtained from Table 1 and Figure 3? I would spend some words on this in the discussion.

Response: We thank the Reviewer for pointing out the missing analysis as well as the redundant table. We have now removed Table 2, instead the information is added to the manuscript text. In addition, we have now added interpretation for the R-DSC AUC measure which evaluates uncertainty quantification in the referral process with multiple thresholds similarly to AUROC measure, and differently from the segmentation performance prediction as reported in Table 1 and Figure 3.

14.Expected entropy results obtained with deep ensemble and MC dropout ensemble are very different in Figure 3, beside being the worst compared to the others. Do you have an explanation for that?

Response: We thank the Reviewer for pointing out this interesting finding. As per previous work, the expected entropy is thought to capture the aleatoric uncertainty e.g., the noise in the data. The results between Deep Ensemble and MC Dropout Ensemble possibly differ due to the latter having larger support of the parameter space due to additional parameter stochasticity via the dropout mechanism, which in turn might identify the uncertainty as mostly parameter uncertainty / mutual information and not as aleatoric / expected entropy. We have now amended the Discussion with this analysis.

15.Line 514: “De Biase et al.”

Response: We thank the Reviewer for pointing out this typo which has now been fixed.

16.Line 518-519: I don't see how using 5 metrics is a limitation.

Response: We thank the Reviewer for pointing out this admittedly modest statement. We added this as we didn't report and reproduce other recently used measures in the literature such as standard entropy and pairwise dice. However, we agree with the reviewer that this does not limit the analysis. We have now revised our limitations paragraph and removed this comment.

Reviewers' comments:

Reviewer #1 (Remarks to the Author):

All my comments have been addressed.

Reviewer #5 (Remarks to the Author):

The authors carefully addressed the comments to the first revision. In addition, they answered to the questions asked and modified the manuscript according to those. Below are few comments to the revised version of the manuscript.

- 1) The goal of the paper is now clear to the reader. However, I believe that the new section dedicated to Uncertainty Estimation in Deep Learning should be included in the Method section instead of the Introduction section. I would modify the Materials and Method section such that it is clear to the reader what is “theory” and what was used to perform the experiments and assess the results in the study. As it is now, I find it hard to see the link between the introduction and this new section which includes formulas. Maybe the section could be located before the dataset.
- 2) In the method is usually used the form $y_{i,j,k}$, when used it for the first time I would specify what i,j,k refer to.

Reviewer #4 (Remarks to the Author):

Sahlsten et al.'s work provides a groundwork for understanding the utility of various, widely used uncertainty quantification methods in the task of OPC GTVp segmentation performed with DL models. The authors methodically assessed five uncertainty measurements for two deep learning models in four experimental settings. The presented DL model performance showed similar results, where comparable, to De Biase et al. and built on this work by systematically assessing various uncertainty metrics. The development of the conditional Dice-risk provided a computationally reasonable method for assessing the sample wise Dice scores. The authors provided clear rationale for each of the uncertainty experiments. Linear regression estimates elucidated the variability of each uncertainty metric alongside DSC, providing insight into the reliability of each metric when the DSC threshold is applied. Exploration of conditional probabilities allowed greater insight into whether a model's degree of certainty in its predictions related to the model's ultimate prediction. Batch referral experiments elucidated the uncertainty metric that most significantly enhanced the model's performance via exclusion of uncertain samples, and instance-based referral showed the clinical utility of each uncertainty metric in relation to a human-in-the-loop segmentation schema. Through these methods and uncertainty measurements, Sahlsten et al. improve the understanding of uncertainty measurements in OPC GTVp segmentation, and sufficiently support their objective of providing a benchmark for future studies.

Comments

The final chosen hyperparameters do not appear to be stated, although the explored parameters were listed in the paragraph beginning on line 180. Please include the final parameters either in text or in a supplemental table.

When describing the uncertainty measures, it would be productive to provide a summary sentence either at the beginning or end of the “Uncertainty measures” section that lists each of the measurements that are going to be used, similar to the sentence that begins on line 433.

Figure 1: Please define theta, p, g, DSCest all on the figure. In general, the focus of the figure should be reconsidered. If it is a data flow schematic then it can be simplified, if it is a computational schematic then it can be improved in terms of definitions and complexity.

Figure 2: Although it's stated in the text and shown by the significance bars in Figure 2, the boxplots are difficult to interpret because they visually show very little to no distinction between the models, despite their being highly significant differences. Could a violin plot or another plot that better captures the data to demonstrate the difference be used in this figure? Otherwise, the boxplot may not even be necessary as it is not visually helpful for the interpretation of the model results.

Figure 3 is hard to read. Needs labels for rows and columns, axes are hard to read, should have unabbreviated titles on axes.

Figure 5: This is nice, but any other appreciable patterns once we've produced these plots? Associations with HPV or PNI (detectable on scans)? Stage or anatomical site differences?

From a readability perspective, this paper would benefit from additional structure (subheadings) in the “Performance evaluation” and “Uncertainty estimation” sections to better delineate the five experiments being performed and to more easily identify the sections of the paper that refer to each of the three experiments. It could also be possible to provide a couple introductory sentences that identify all the experiments before going into the details of the experiments that require additional description. In these sentences, the full description of the linear regression model could be provided, given that it is short and does not necessarily need its own subsection in the “Performance evaluation” section. More clearly delineating the experiments and each of their results would improve the reader's ability to easily orient themselves to the conditions of each experiment and better interpret results without having to return to the “Performance evaluation” section. Furthermore, the intended audience should be considered. For example, I'm not sure the logical function "V" will be familiar to Communications Medicine readership without a framing sentence.

When predictive entropy and mutual information are introduced in line 224, it would be helpful to identify the abbreviations that are later used to refer to these variables in parentheses, i.e. predictive entropy (UE) and mutual information (UMI) because these are the only two uncertainty measures that are not referred to in their formal definitions by the variable names that they are referred to in the tables listed in the results section. While the scatterplot does list the names of each of the uncertainty variables, it would be beneficial to declare these variables earlier so that one is not scouring the figure legends for definitions of variables that have already been defined earlier in the text but referred to under a different variable name.

When using a specific clinical context such as OP-HNSCC, it's important not to ignore relevant factors within that data structure. Additional descriptive and demographic statistics are needed for the data set. What is the proportion of HPV+ cancer? What is the proportion of tonsil vs. base of tongue primary sites? Performance should be shown to be equal over these uniformly recognized cancer subclasses.

Can a simultaneous evaluation of uncertainty and performance be done on a continuous scale instead of thresholding? I think readership would find visualization of this relationship interesting. Similarly, did authors consider selecting UQ threshold values based on optimizing a final metric instead of pre-defining points of DSC (such as Dolezal Nature Communications 2022)?”

Specific Reviewer Comments and Author's reply

We would like to thank the editors and reviewers of Communications Medicine for thorough reading our manuscript and providing valuable suggestions. We trust to have addressed all the reviewers' concerns, and improved the manuscript accordingly. Reviewers' comments are listed below, with our corresponding responses annotated in blue. All corrections and additions are highlighted in the edited manuscript file in yellow, green, and cyan corresponding to responses to Reviewer #4, Reviewer #5, or both, respectively. Note that Reviewer #4 remarks based on the original manuscript, not on the revised manuscript. Thus, we also highlight prior changes based on the first revision round Reviewers' comments.

Reviewer #1 (Remarks to the Author):

All my comments have been addressed.

Response: We thank the Reviewer for this acknowledgment.

Reviewer #5 (Remarks to the Author):

The authors carefully addressed the comments to the first revision. In addition, they answered to the questions asked and modified the manuscript according to those. Below are few comments to the revised version of the manuscript.

1) The goal of the paper is now clear to the reader. However, I believe that the new section dedicated to Uncertainty Estimation in Deep Learning should be included in the Method section instead of the Introduction section. I would modify the Materials and Method section such that it is clear to the reader what is “theory” and what was used to perform the experiments and assess the results in the study. As it is now, I find it hard to see the link between the introduction and this new section which includes formulas. Maybe the section could be located before the dataset.

Response: We thank the reviewer for this comment and recommendation to the revised manuscript. We agree that the theory of uncertainty estimation in deep learning is better suited for the Methods section and it has now been moved to the Methods section as suggested. In addition, the Methods section now has two subsections i.e., “Uncertainty Estimation in Deep Learning” and “Materials and Experimental Setup”, which clearly differentiate between the theory and our experiments.

2) In the method is usually used the form $y_{i,j,k}$, when used it for the first time I would specify what i,j,k refer to.

Response: We thank the reviewer for the comment. We have now added a description of i,j,k following their initial use.

Reviewer #4 (Remarks to the Author, to the original manuscript):

Sahlsten et al.'s work provides a groundwork for understanding the utility of various, widely used uncertainty quantification methods in the task of OPC GTVp segmentation performed with DL models. The authors methodically assessed five uncertainty measurements for two deep learning models in four experimental settings. The presented DL model performance showed similar results, where comparable, to De Biase et al. and built on this work by systematically assessing various uncertainty metrics. The development of the conditional Dice-risk provided a computationally reasonable method for assessing the sample wise Dice scores. The authors provided clear rationale for each of the uncertainty experiments. Linear regression estimates elucidated the variability of each uncertainty metric alongside DSC, providing insight into the reliability of each metric when the DSC threshold is applied. Exploration of conditional probabilities allowed greater insight into whether a model's degree of certainty in its predictions related to the model's ultimate prediction. Batch referral experiments elucidated the uncertainty metric that most significantly enhanced the model's performance via exclusion of uncertain samples, and instance-based referral showed the clinical utility of each uncertainty metric in relation to a human-in-the-loop segmentation schema. Through these methods and uncertainty measurements, Sahlsten et al. improve the understanding of uncertainty measurements in OPC GTVp segmentation, and sufficiently support their objective of providing a benchmark for future studies.

Response: We thank the reviewer for in depth reading of our original manuscript and seeing its value and enhancing the understanding of uncertainty quantification in OPC GTVp segmentation with DL models.

Comments

The final chosen hyperparameters do not appear to be stated, although the explored parameters were listed in the paragraph beginning on line 180. Please include the final parameters either in text or in a supplemental table.

Response: We thank the reviewer for this comment. However, this point was raised by another Reviewer from the original manuscript which was accordingly revised. Thus, the hyperparameters are found in the "Bayesian deep learning models" subsection.

When describing the uncertainty measures, it would be productive to provide a summary sentence either at the beginning or end of the "Uncertainty measures" section that lists each of the measurements that are going to be used, similar to the sentence that begins on line 433.

Response: We thank the reviewer for this comment. However, similar points about the lack of clarity about the "Uncertainty measures" subsection were already raised by the other Reviewers from the original manuscript which we accordingly revised such that the used uncertainty measures were clearly listed.

Figure 1: Please define θ , p , g , DSC_{est} all on the figure. In general, the focus of the figure should be reconsidered. If it is a data flow schematic then it can be simplified, if it is a computational schematic then it can be improved in terms of definitions and complexity.

Response: We thank the reviewer for pointing out missing definitions and improper focus in Figure 1. We have clarified and simplified Figure 1 which now matches the experimental design

of the manuscript better. Moreover, the number of abbreviations was reduced and they are now defined in the figure caption.

Figure 2: Although it's stated in the text and shown by the significance bars in Figure 2, the boxplots are difficult to interpret because they visually show very little to no distinction between the models, despite their being highly significant differences. Could a violin plot or another plot that better captures the data to demonstrate the difference be used in this figure? Otherwise, the boxplot may not even be necessary as it is not visually helpful for the interpretation of the model results.

Response: We thank the reviewer for pointing out this lack of visual clarity. We have now changed the visualization of Figure 2 to a violinplot as suggested, which highlights the differences slightly better. However, the main focus of the figure is to show the overall performance of the models before uncertainty quantification. Thus, we decided to keep this new figure.

Figure 3 is hard to read. Needs labels for rows and columns, axes are hard to read, should have un-abbreviated titles on axes.

Response: We thank the reviewer for this helpful comment. However, a similar point was raised by another Reviewer from the original manuscript which was accordingly revised. The labels and legends of Figure 3 in the revised manuscript are with larger font sizes. We decided to keep the abbreviated titles on axes, since the uncertainty measures have lengthy names that would need to be written with small font size to fit the panels of the plots. The abbreviations are clearly defined in the figure caption. Note that the x-axis is shared between columns, which is now clarified in the figure caption.

Figure 5: This is nice, but any other appreciable patterns once we've produced these plots? Associations with HPV or PNI (detectable on scans)? Stage or anatomical site differences?

Response: As noted in a comment below, we have added an additional supplementary table in Appendix A (Table A2) containing clinically relevant information about the external validation set, such as statistics and frequencies regarding the patients' age, gender, tumor subsite, and TMC staging. Notably, all patients in the external validation set were HPV+, so stratified analysis using HPV status was not possible in our study. To aid in reader interpretation of uncertainty map plots (both in the main text and Appendix), we have added text to give additional clinical context to the cases visualized. Upon visual inspection, we did not appreciate any discernible patterns in the visualization of different subsets of cases. However, we believe future work should systematically investigate quantitative and qualitative differences in voxel-level uncertainty maps based on subgroups with a particular focus on HPV status. We have added this suggested future direction in the Discussion section.

From a readability perspective, this paper would benefit from additional structure (subheadings) in the "Performance evaluation" and "Uncertainty estimation" sections to better delineate the five experiments being performed and to more easily identify the sections of the paper that refer to each of the three experiments. It could also be possible to provide a couple introductory sentences that identify all the experiments before going into the details of the experiments that require additional description. In these sentences, the full description of the linear regression model could be provided, given that it is short and does not necessarily need its own subsection in the "Performance evaluation" section. More clearly delineating the experiments and each of their results would improve the reader's ability to easily orient

themselves to the conditions of each experiment and better interpret results without having to return to the “Performance evaluation” section. Furthermore, the intended audience should be considered. For example, I’m not sure the logical function “V” will be familiar to Communications Medicine readership without a framing sentence.

Response: We thank the reviewer for this comment. However, this point was raised by another Reviewer from the original manuscript which was accordingly revised. Specifically the Methods section was clarified and structured to include five distinct subsections i.e., the Dataset, Bayesian deep learning models, Segmentation performance evaluation, Uncertainty measures, and Uncertainty performance evaluation. In the revised version, the three different experiments were listed in the “Uncertainty performance evaluation” subsection, while all evaluated uncertainty measures were listed in the “Uncertainty measures” subsection as well as in an introductory paragraph in “Introductions” section. In regards to the final comment, we assumed that the “V” logical function remark refers to the AvU (Accuracy vs Uncertainty) measure that was misinterpreted as a logical operation but actually is a measure of accuracy. The description for this measure was already clarified in the revised manuscript as well as abbreviated as AU to reduce ambiguity.

When predictive entropy and mutual information are introduced in line 224, it would be helpful to identify the abbreviations that are later used to refer to these variables in parentheses, i.e. predictive entropy (UE) and mutual information (UMI) because these are the only two uncertainty measures that are not referred to in their formal definitions by the variable names that they are referred to in the tables listed in the results section. While the scatterplot does list the names of each of the uncertainty variables, it would be beneficial to declare these variables earlier so that one is not scouring the figure legends for definitions of variables that have already been defined earlier in the text but referred to under a different variable name.

Response: We thank the reviewer for this comment. However, this remark was also raised by another Reviewer of the original manuscript and accordingly revised. In the revised manuscript all notations for uncertainty measures were standardized. Specifically the information theoretic and previously proposed measures follow the literature notation i.e., predictive entropy is now H , mutual information is now I , coefficient of variation is now CV , and the expected entropy is now EH . Moreover, our proposed risk based measure is now R_{DSC} . The structure variants for the entropy measures are prefixed with letter S, i.e., structure predictive entropy is now SH , structure mutual information is now SI , and structure expected entropy is now SEH .

When using a specific clinical context such as OP-HNSCC, it's important not to ignore relevant factors within that data structure. Additional descriptive and demographic statistics are needed for the data set. What is the proportion of HPV+ cancer? What is the proportion of tonsil vs. base of tongue primary sites? Performance should be shown to be equal over these uniformly recognized cancer subclasses.

Response: We thank the reviewer for raising the clinically relevant point. We completely agree that these additional details on clinical and demographic variables should be made known to the reader, particularly for the external validation set (the corresponding information for the HECKTOR training set is already referenced in its corresponding publication). Therefore, we have added an additional supplementary table in Appendix A (Table A2) containing clinically relevant information about the external validation set, such as statistics and frequencies regarding the patients’ age, gender, tumor subsite, and TMC staging. We have now explicitly referenced

where readers can find this supplementary information in the Methods section of the main text. Notably, all patients in the external validation set were HPV+, which has now been explicitly mentioned as well. Regarding equality of performance, due to the relatively small size of our external test set, additional stratified analyses were deemed to be outside the scope of the current work. However, we do believe this is an important area for future investigation and have therefore added comments on this topic to our Discussion section.

Can a simultaneous evaluation of uncertainty and performance be done on a continuous scale instead of thresholding? I think readership would find visualization of this relationship interesting. Similarly, did authors consider selecting UQ threshold values based on optimizing a final metric instead of pre-defining points of DSC (such as Dolezal Nature Communications 2022)?”

Response: We thank the reviewer for this comment. We have now added a linear regression analysis with uncertainty values being the independent values and segmentation performance, in terms of DSC, being the dependent values to Appendix B in order to visualize simultaneous uncertainty and performance evaluation on a continuous scale. In short the analysis showed similar outcomes to our prior analysis i.e., coefficient of variation having the strongest Pearson correlation coefficient, which we deemed as a suitable evaluation measure for this experiment. We decided not to include UQ threshold optimization in terms of the final metric as we focused our UQ analysis mainly on the patient-level uncertainty instead of tile-based uncertainty, the latter of which is not available for all of the evaluated uncertainty measures such as DSC-risk and coefficient of variation.

REVIEWERS' COMMENTS:

Reviewer #4 (Remarks to the Author):

The authors have made significant and important improvements to the manuscript. In particular, I appreciate the reformatting of methods, the improvement in figure readability, and the additional supplemental information.

Reviewer #5 (Remarks to the Author):

All my comments have been addressed.

Specific Reviewer Comments and Author's reply

We would like to thank the editor and reviewers of Communications Medicine for thorough reading our manuscript. Our response to the reviewers' remarks are attached below.

Reviewer #4 (Remarks to the Author):

The authors have made significant and important improvements to the manuscript. In particular, I appreciate the reformatting of methods, the improvement in figure readability, and the additional supplemental information.

Authors' response: We thank the reviewer for the positive remarks.

Reviewer #5 (Remarks to the Author):

All my comments have been addressed.

Authors' response: We thank the reviewer for the positive remark.